# Differentially Private Conformal Prediction via Quantile Binary Search

**Ogonnaya M. Romanus**  *romanusogonnaya@gmail.com*
*Department of Mathematics and Statistics*
*Auburn University*

**Roberto Molinari**  *robmolinari@auburn.edu*
*Department of Mathematics and Statistics*
*Auburn University*

Reviewed on OpenReview: *https://openreview.net/forum?id=IK7tNOucJ3*

## Abstract

Differentially Private (DP) approaches have been widely explored and implemented for a broad variety of tasks delivering corresponding privacy guarantees in these settings. While most of these DP approaches focus on limiting privacy leakage from training data, there are fewer approaches that consider leakage when procedures involve *calibration data* which is common in uncertainty quantification through Conformal Prediction (CP). Since there is a limited amount of approaches in this direction, in this work we deliver a general DP approach for CP that we call Private Conformity via Quantile Search (P-COQS). The proposed approach adapts an existing randomized binary search algorithm for computing DP quantiles in the calibration phase of CP thereby guaranteeing privacy of the consequent prediction sets. This however comes at a price of marginally under-covering with respect to the desired $(1 - \alpha)$-level when using finite-sample calibration sets (although broad empirical results show that the P-COQS generally targets the required level in the considered cases). Confirming properties of the adapted algorithm and quantifying the approximate coverage guarantees of the consequent CP, we conduct extensive experiments to examine the effects of privacy noise, sample size and significance level on the performance of P-COQS compared to existing alternatives. In addition, we empirically evaluate our approach on several benchmark datasets, including CIFAR-10, ImageNet and CoronaHack. Our results suggest that the proposed method is robust to privacy noise and performs favorably with respect to the current DP alternative in terms of *empirical coverage*, *efficiency*, and *informativeness*. Specifically, the results indicate that P-COQS produces smaller conformal prediction sets while simultaneously targeting the desired coverage and privacy guarantees in all these experimental settings.

# 1 Introduction

In high-stakes domains such as health care, criminal justice and financial risk assessment (among others) machine learning models are increasingly being used to guide decision-making processes. These approaches are commonly built to produce single-point predictions without quantifying uncertainty, which could lead to overconfident and unreliable outcomes with negative consequences for the following decisions. Many techniques have been proposed to address the need for reliable uncertainty quantification, including (but not limited to) Bayesian neural networks, ensemble methods, and dropout-based approximations (see e.g. Blundell et al., 2015; Lakshminarayanan et al., 2017; Osband et al., 2016; Gal & Ghahramani, 2016; Kendall & Gal, 2017). While these methods can offer meaningful uncertainty estimates, they often rely on strong modeling assumptions, are computationally intensive, or lack rigorous coverage guarantees (especially under model misspecification or when applied out-of-distribution). Conformal Prediction (CP) addresses various of these limitations by generating *prediction sets*, that is, subsets of the response space $\mathcal{Y}$ that, under the assumption of exchangeability, are guaranteed to contain the true response with a user-specified probability $1 - \alpha$, where $\alpha \in (0, 1)$. These sets provide a reliable and model-agnostic framework for uncertainty quantification, making them increasingly attractive in safety-critical applications (see e.g. Uddin & Lofstrom, 2023; Astigarraga et al., 2025; Tuwani & Beam, 2023; Olsson et al., 2022). While conformal methods require independent and identically distributed (iid) data for exact guarantees, recent extensions have begun to explore how to adapt them to distribution shift settings with approximate validity (see e.g. Tibshirani et al., 2019; Gibbs & Candès, 2021; Jonkers et al., 2024).

However, while CP provides statistical reliability, its reliance on calibration data introduces privacy risks since any publicly released statistic based on these data could be exploited by an adversary. Moreover, when training models on data that collect individual-level information, these models tend to preserve individual-specific information, and some of them, such as deep neural networks, are known to *memorize* training examples (Fredrikson et al., 2015; Carlini et al., 2021), enabling adversaries to extract sensitive information via model inversion (Fredrikson et al., 2015) or membership inference attacks (Shokri et al., 2017). This poses a significant challenge also in regulated domains (e.g., under GDPR, HIPAA, or CCPA), where models must not leak information about individuals in the training or calibration sets. To address these issues, Differential Privacy (DP) (Dwork, 2006) has emerged as one of the main frameworks for privacy-preserving machine-learning, ensuring (with high probability) that model outputs do not reveal whether any individual's data was included in the training set. However, existing DP frameworks focus primarily on protecting the *training* phase, leaving the calibration phase (and thus the CP step) vulnerable to privacy breaches. Given this issue, there are nevertheless very few existing approaches that ensure that CP procedures can provide DP guarantees while maintaining reasonable performance in terms of uncertainty quantification metrics and computational efficiency (see Angelopoulos et al., 2022). Hence, this work aims to further contribute to the goal of uncertainty quantification under privacy constraints by adapting an existing DP quantile selection approach based on binary search (Huang et al., 2021) and using it as a subroutine to deliver CP sets that guarantee privacy. We confirm the theoretical properties of the resulting CP procedure and highlight its favorable experimental performance compared to the few existing alternatives, both in terms of uncertainty quantification and computational efficiency.

## 2 Related Work

The idea behind CP was first described in Vovk et al. (2005) followed by different papers that developed its theory and applications (see Vovk (2015), Vovk (2002), Vovk et al. (2017), Vovk & Petej (2012), and the references therein). Based on these, considerable attention has been paid in recent years to adapt CP to different settings and deliver additional applied and theoretical results (see Angelopoulos & Bates (2021) for a recent detailed review). Despite these different research outputs, little attention has been devoted to guaranteeing privacy for the calibration data through protection of the derived *non-conformity scores* used to deliver the prediction sets for uncertainty quantification. To the best of the authors' knowledge, the only existing work related to the central DP approach we present in this paper is that of Angelopoulos et al. (2022) where they privatize the CP procedure by discretizing the non-conformity scores into bins and applying the *exponential mechanism* (McSherry & Talwar, 2007) to select a DP quantile which is then used to form the prediction sets. More specifically, to maintain coverage guarantees under $\epsilon$-DP, the target quantile is adjusted (inflated) to account for added noise. This ensures both privacy (via DP) and valid coverage, albeit with potentially larger prediction sets due to the privacy-utility trade-off. A possible practical limitation of this proposed approach lies in the choice of the optimal number of bins, as well as a hyperparameter that needs to be tuned to minimize the inflated quantile to achieve the smallest possible prediction set. Although procedures have been determined to compute these in an optimal manner, they can potentially increase the time complexity of this method. In recent years there have been few other CP approaches proposed in the setting of federated learning (Humbert et al., 2023) and *local DP* (Penso et al., 2025), however, these do not fit in the central DP framework of this paper.

## 3 Preliminaries

### 3.1 Conformal Prediction

To define the framework of the proposed method, let us consider $n$ exchangeable data points $\mathcal{D} = \{(X_1, Y_1), \ldots, (X_n, Y_n)\}$ with $X_i \in \mathcal{X}^d$ being a $d$-dimensional input vector and $Y_i \in \mathcal{Y}$ being the target variable. Using the basic *split* CP (Papadopoulos et al., 2002; Lei et al., 2015), we can partition these points into a training set with $n_{\text{train}}$ observations (to fit a model of choice $f$) and a calibration set with $n_{\text{cal}} = n - n_{\text{train}}$ observations to quantify prediction uncertainty (assuming test data is taken from another sample). If we let $\mathcal{I} = \{1, \ldots, n\}$ represent the index set of the observations, then we can define $\mathcal{I}_{\text{cal}} \subset \mathcal{I}$ as the index set of observations in the calibration set. Assuming that we have trained a model $\hat{f}$ on the training set, the goal of CP is to compute *non-conformity scores* $s_i = s(X_i, Y_i, \hat{f})$ for each calibration point $(X_i, Y_i), \forall i \in \mathcal{I}_{\text{cal}}$. These scores measure the discrepancy between the model's prediction and the true target value such that lower scores suggest better agreement, that is, better predictions from the trained model $\hat{f}$. Therefore we now have a set of non-conformity scores $\mathcal{S} = \{s_1, \ldots, s_{\text{cal}}\}$ on which we can obtain $\alpha$-quantiles with $\alpha \in (0, 1)$ being the significance level which represents the amount of errors (in percentage) that the model is allowed to make. We denote this quantile as $q$ and let its dependence on the significance level $\alpha$ be implicit. Once the scores are computed, we can therefore sort them in increasing order (that is, $\{s_{[1]} \leq \ldots \leq s_{[n_{\text{cal}}]}\}$) and estimate this quantile by selecting the score in the $r^{\text{th}}$ position where we define $r = \lceil (1 - \alpha)(n_{\text{cal}} + 1) \rceil$, with $\lceil v \rceil$ representing the smallest integer greater than $x$, leading to $\hat{q} = s_{[r]}$. Suppose that we now have a new test data point $X_{\text{test}}$, the resulting prediction

set for the target $Y_{\text{test}}$ is defined as:

$$\mathcal{C}(X_{\text{test}}) = \left\{ Y \in \mathcal{Y} : s(X_{\text{test}}, Y, \hat{f}) \leq \hat{q} \right\}.$$

Under the assumption of exchangeability of the data points, this set guarantees the coverage probability $\mathbb{P}(Y_{\text{test}} \in \mathcal{C}(X_{\text{test}})) \geq 1 - \alpha$, where the probability is taken over the $n + 1$ data points $(X_1, Y_1), \ldots, (X_n, Y_n), (X_{\text{test}}, Y_{\text{test}})$. For further details and extensions we refer the reader to Angelopoulos & Bates (2021).

## 3.2 Differential Privacy

Differential Privacy (DP) currently represents one of the main frameworks for privacy protection with mathematically provable guarantees. More in detail, a randomized mechanism $\mathcal{M}$ is said to be $(\epsilon, \delta)$-DP if, for all neighboring datasets $\mathcal{D}$ and $\mathcal{D}'$ (that is, datasets with Hamming distance $\Delta(\mathcal{D}, \mathcal{D}') = 1$) and for all measurable subsets $\mathcal{Q}$ of the output space, we have that

$$\mathbb{P}[\mathcal{M}(\mathcal{D}) \in \mathcal{Q}] \leq e^{\epsilon} \mathbb{P}[\mathcal{M}(\mathcal{D}') \in \mathcal{Q}] + \delta.$$

There are many variations of this framework, including Gaussian Differential Privacy which benefits from a series of nice additional properties compared to the above $(\epsilon, \delta)$-DP (Dwork, 2006; Dong et al., 2022). For the purpose of this work, we rely on a specific version of DP defined below.

**Definition 1** (Zero-Concentrated Differential Privacy, Bun & Steinke (2016)). *For $\rho > 0$, a randomized mechanism $\mathcal{M}$ satisfies $(\rho, 0)$-Zero-Concentrated Differential Privacy (zCDP) if for all neighboring datasets $\mathcal{D}$ and $\mathcal{D}'$ and for all $\eta > 1$, the Rényi divergence $D$ of order $\eta$ satisfies:*

$$D_\eta(\mathcal{M}(\mathcal{D}) \| \mathcal{M}(\mathcal{D}')) \leq \rho \, \eta.$$

Unlike $(\epsilon, \delta)$-DP, $\rho$-zCDP is *tight* under sequential composition without the need for the *advanced composition theorem*, which may consume extra privacy budget. Nevertheless, there is a direct relationship between the $(\epsilon, \delta)$-DP framework and the zCDP framework: indeed, if $\mathcal{M}$ satisfies $\rho$-zCDP, then for $\delta > 0$, $\mathcal{M}$ satisfies $(\epsilon, \delta)$-DP with $\epsilon = \rho + 2\sqrt{\rho \log(1/\delta)}$ (see Near & Abuah, 2021). Hence, the latter implies the former. In particular, a randomized mechanism $\mathcal{M}$ satisfies $(\epsilon, 0)$-DP if and only if it satisfies $\epsilon$-zCDP (Bun & Steinke, 2016).

## 4 P-COQS: Private Conformity via Quantile Search

As highlighted in Angelopoulos et al. (2022), the release of the non-conformity score quantile $\hat{q}$ or of the prediction sets based on this quantile can compromise the privacy of individuals in the calibration set. Therefore, a solution to this problem is to directly release a DP quantile or its corresponding prediction sets, which would also preserve DP based on the privatized quantile. While the existing alternative in Angelopoulos et al. (2022) employs the exponential mechanism to sample a DP quantile, this work relies on a direct adaptation of the DP binary search algorithm presented in Huang et al. (2021). More specifically, the latter algorithm releases a DP quantile by employing a noisy count function (denoted as `NoisyRC`) within a binary search procedure run on a sequence of ordered integers. In particular, the function `NoisyRC`$([a, b], \mathcal{G})$ returns a noisy count of $\mathcal{G} \cap [a, b]$, where $\mathcal{G}$ is a set containing ordered integers that are considered sensitive and hence must

be accessed privately (see Huang et al., 2021). However, since in this work our sequences are not integers (given that the non-conformity scores are continuous), we adapt this DP quantile selection approach in a straightforward manner and, for completeness, we reproduce their adapted approach in Algorithm 1 with some modification to suit our application.

---

**Algorithm 1** Adapted `PrivQuant` Algorithm of Huang et al. (2021)

---

**Require:** Non-conformity scores $\mathcal{S} = \{s_1, \ldots, s_{n_{\mathrm{cal}}}\}$, significance level $\alpha \in (0, 1)$, lower and upper bounds for non-conformity scores $[a, b]$, $\Delta > 0$ (small positive value, default $\Delta = 10^{-10}$), privacy parameter $\rho$

**Ensure:** DP quantile $q^{\mathrm{DP}}$

 1: Fix $r = \lceil (1 - \alpha)(n_{\mathrm{cal}} + 1) \rceil$ and $N = \lceil \log_2 \left( {}^{b-a}/\Delta \right) \rceil$
 2: Initialize: left $\leftarrow a$, right $\leftarrow b$, $i \leftarrow 0$
 3: **while** $i < N$ **do**
 4:      mid $\leftarrow \frac{\text{left}+\text{right}}{2}$
 5:      $c \leftarrow \mathtt{NoisyRC}([a, \text{mid}], \mathcal{S})$
 6:      **if** $c < r$ **then**
 7:          left $\leftarrow$ mid $+ \Delta$
 8:      **else**
 9:          right $\leftarrow$ mid
10:      **end if**
11:      $i \leftarrow i + 1$
12: **end while**
13: **return** $q^{\mathrm{DP}} = \frac{\text{left}+\text{right}}{2}$

---

Broadly speaking, Algorithm 1 performs a binary search over the interval $[a, b]$ (often corresponding to $[0, 1]$ in the case of classification tasks) by iteratively narrowing the search space to localize the quantile of interest while preserving privacy through a randomized counting mechanism. More specifically, at each iteration, the algorithm evaluates a midpoint and calls the function `NoisyRC`, which returns a DP count of the number of non-conformity scores (in $\mathcal{S}$) less than or equal to the midpoint. This count guides the search by indicating whether the true quantile lies to the left or right of the midpoint. The process continues until it reaches iteration $N$ which is the maximum number of iterations allowed to ensure $\rho$-zCDP. Indeed, as stated further on, the DP noise used for the function `NoisyRC` needs to be scaled to the number of iterations in Algorithm 1. Since we would ideally want the algorithm to terminate when the interval length is smaller than a small value $\Delta$ (i.e. right $-$ left $\leq \Delta$), the fixed number of iterations $N$ needed to reach this point is such that ${}^{b-a}/2^N \leq \Delta$ (since the interval is divided by two after each iteration) which, solving for $N$, gives us

$$N = \lceil \log_2 \left( {}^{b-a}/\Delta \right) \rceil.$$

With this representing the exact number of iterations to guarantee the required condition, Algorithm 1 returns the midpoint as the private quantile estimate $q^{\mathrm{DP}}$.

**Remark 1.** *As for all DP settings, the choice of the bounds $[a, b]$ have to either be made a-priori (before seeing the data) or also have to be determined through DP approaches. For classification problems, the bounds for non-conformity scores are naturally defined (in general) on the $[0, 1]$ space. In regression settings however, non-conformity scores such as absolute or squared residuals are unbounded, and thus a finite range $[a, b]$ must be imposed. A common and privacy-safe choice is to*

*fix $a = 0$ and truncate scores above a robust upper bound $b$ estimated through a differentially private procedure with a small additional privacy budget $\rho_b$. In the latter case the total privacy cost then becomes $\rho_{total} = \rho + \rho_b$. In practice, using a conservative public bound or a privately estimated high quantile yields similar utility while preserving the formal $\rho_{total}$-zCDP guarantee.*

Once this quantile is obtained, following the definition in Section 3.1, the DP conformal prediction set is simply given by:

$$\mathcal{C}^{DP}(X_{\text{test}}) = \left\{ Y \in \mathcal{Y} : s(X_{\text{test}}, Y, \hat{f}) \leq q^{DP} \right\}. \tag{1}$$

We refer to the above procedure as Private Conformity via Quantile Search (P-COQS) which therefore is built on the adaptation described in Algorithm 1 and the consequent use of the resulting DP quantile $q^{DP}$ in equation 1. The entire P-COQS procedure is summarized in Algorithm 2.

---

**Algorithm 2** P-COQS

---

**Require:** Training data $\mathcal{D}_{\text{train}} = \{(X_1, Y_1), \ldots, (X_{n_{\text{train}}}, Y_{n_{\text{train}}})\}$, calibration data $\mathcal{D}_{\text{cal}} = \{(X_1, Y_1), \ldots, (X_{n_{\text{cal}}}, Y_{n_{\text{cal}}})\}$, test point $X_{\text{test}}$, significance level $\alpha \in (0, 1)$, bounds $[a, b]$, resolution $\Delta > 0$, privacy parameter $\rho$
**Ensure:** DP prediction set $\mathcal{C}^{\text{DP}}$
1: Train model on $\mathcal{D}_{\text{train}}$ to obtain $\hat{f}$
2: Compute non-conformity scores $\mathcal{S} = \{s_1, \ldots, s_{n_{\text{cal}}}\}$ from $\mathcal{D}_{\text{cal}}$ using $\hat{f}$
3: Run Algorithm 1 to obtain $q^{DP}$
4: Define $\mathcal{C}^{DP}(X_{\text{test}}) = \{Y \in \mathcal{Y} : s(X_{\text{test}}, Y, \hat{f}) \leq q^{DP}\}$
5: **return** $\mathcal{C}^{DP}$

---

Let us now focus on the properties of P-COQS and, for completeness, we start by studying the error bound and DP properties of Algorithm 1 which directly follow from the properties of the `PrivQuant` algorithm in Huang et al. (2021). Indeed, while the latter algorithm is tailored to search over positive integers, as mentioned earlier, Algorithm 1 adapts the latter to any ordered data type and any set of bounds $[a, b]$ on the real line. Let us therefore discuss (confirm) the DP properties of this adaptation and, to do so, let us first define the noisy count function as

$$\texttt{NoisyRC}([a, \text{mid}], \mathcal{S}) = \texttt{card}\big(\mathcal{S} \cap [a, \text{mid}]\big) + \mathcal{N}\left(0, \frac{\lceil \log_2^{(b-a/\Delta)} \rceil}{2\rho}\right), \tag{2}$$

where $\rho > 0$ is the privacy parameter under zCDP, with the sensitivity scaled to the maximum number of iterations $N$ for Algorithm 1 (considering that count query $l_1$-sensitivity is 1), and `card` denotes the cardinality of set. Let us now denote $u = {}^{b-a}/\Delta$ and let $\Phi$ represent the standard normal CDF. Moreover, let $M(\Delta) = \max_t \#\{ s_i \in (t, t + \Delta] \}$ represent the maximum number of calibration scores inside a $\Delta$-interval. Then, adapting directly from Huang et al. (2021), the following proposition holds.

**Proposition 1.** *Algorithm 1 using equation 2 is $\rho$-zCDP and, with probability at least $1 - \beta$, returns a quantile whose rank error is bounded by*

$$\tau \leq \tau^* + M(\Delta), \quad \text{where} \quad \tau^* = \sqrt{\frac{\lceil \log_2(u) \rceil}{\rho} \log\left(\frac{2\lceil \log_2(u) \rceil}{\beta}\right)}.$$

*Proof.* Since the sensitivity of the range query $\texttt{card}(\mathcal{S} \cap [a, \text{mid}])$ is 1, adding Gaussian noise $\mathcal{N}\left(0, \frac{N}{2\rho}\right)$ guarantees $\rho/N$-zCDP per call, where $N = \lceil \log_2(u) \rceil$. By composition (Bun & Steinke, 2016), after $N$ calls the whole algorithm satisfies $\rho$-zCDP. Now, let $W \sim \mathcal{N}(0, \sigma^2)$ with $\sigma^2 = N/(2\rho)$. Then the error between the true count and its noisy version is distributed as $W$. For one call we have that

$$\Pr\left(|W| > \tau^*\right) \leq 2e^{-\tau^{*2}/(2\sigma^2)}.$$

Applying a union bound over $N$ calls we consequently have

$$\Pr\left(\exists i : |W_i| > \tau^*\right) \leq 2Ne^{-\tau^{*2}/(2\sigma^2)}.$$

To ensure this probability is at most $\beta$, it suffices that

$$\tau^* \geq \sigma \sqrt{2 \log\left(\frac{2N}{\beta}\right)}.$$

Finally, since the algorithm stops with an interval of length at most $\Delta$, the true count can vary by at most $M(\Delta)$ within this interval so, substituting $\sigma^2 = N/(2\rho)$ and $N = \lceil \log_2(u) \rceil$, the final rank error is at most $\tau^* + M(\Delta)$. $\qquad\square$

**Remark 2.** *If the score distribution admits a density bounded by $L$ on $[a, b]$, then we have that $\mathbb{E}[M(\Delta)] \leq L \Delta n_{\text{cal}}$. Consequently, choosing $\Delta \asymp c/n_{\text{cal}}$ implies that $\mathbb{E}[M(\Delta)] \lesssim cL$, so that the dominant term in the rank error becomes $\tau^*$. Hence, smaller $\Delta$ increases $N = \lceil \log_2((b-a)/\Delta) \rceil$ (and thus $\tau^*$), while larger $\Delta$ increases $M(\Delta)$. Consequently, $\Delta$ should balance the trade-off between the DP-noise component ($\tau^* \propto \sqrt{\log(1/\Delta)}$) and the discretization component ($M(\Delta) \propto \Delta n_{\text{cal}}$ in expectation). However, in contrast to other possible DP quantile approaches which use histograms (and therefore require careful tuning of a global bin width), the choice of $\Delta$ here is considerably simpler. Indeed, $\Delta$ only controls the numerical resolution of the binary search and does not define a partition of the score space. As a result, any $\Delta$ in the regime $\Delta \asymp c/n_{\text{cal}}$ leads to similar behavior, making the tuning of $\Delta$ substantially easier than selecting an "optimal" bin size in histogram-based approaches.*

**Remark 3.** *With respect to the DP noise error term $\tau^*$, an equivalent (but slightly sharper) formulation avoids the union bound and uses the Gaussian quantile. In that case,*

$$\tau^* = \sigma \, \Phi^{-1}\left(1 - \frac{\beta}{2N}\right) = \sqrt{\frac{\lceil \log_2(u) \rceil}{2\rho}} \, \Phi^{-1}\left(1 - \frac{\beta}{2\lceil \log_2(u) \rceil}\right).$$

**Remark 4.** *Noting that $\lceil \log_2 u \rceil \leq \log_2 u + 1$, we have that*

$$\tau^* \leq \sqrt{\frac{\log_2 u + 1}{\rho} \log\left(\frac{2(\log_2 u + 1)}{\beta}\right)} = \sqrt{\frac{\log u}{\rho \log 2} \left(\log \log u + O(1)\right)}.$$

*Therefore, with $\rho, \beta$ fixed, as $u \to \infty$ (i.e. as $\Delta \to 0$) we have that*

$$\tau^* = \Theta\left(\sqrt{\frac{\log u \, \log \log u}{\rho}}\right),$$

*which matches the polylogarithmic order in Huang et al. (2021) (near-optimal up to constants).*

**Remark 5.** *A natural question is whether empirical membership-inference attacks (MIAs) can be used to validate privacy leakage from releasing a DP quantile. Since P-COQS relies on the adapted PrivQuant mechanism in Algorithm 1, which satisfies $\rho$-zCDP and therefore inherits explicit $(\varepsilon, \delta)$-DP guarantees, the worst-case success probability of any MIA is already upper-bounded by the theoretical DP conversion bounds (Yeom et al., 2018). Thus, MIAs cannot reveal leakage beyond what the DP guarantee already certifies. Nevertheless, MIAs can serve as an empirical diagnostic to confirm that the observed attack advantage is consistent with the theoretical bound, and therefore may be useful in high-risk practical deployments. This investigation is however left for future work.*

Appendix E provides some empirical results on the rank errors of Algorithm 1 based on the settings presented later in Section 5.1. Following the above, and for sake of completeness, we also provide the computational complexity of this algorithm.

**Proposition 2.** *Algorithm 1 runs in*

$$\mathcal{O}\left(n_{\text{cal}} \log\left(\frac{b-a}{\Delta}\right)\right)$$

*time and requires $\mathcal{O}(1)$ memory beyond the storage of the non-conformity scores.*

Proposition 2 is a direct consequence of the binary search routine over $[a, b]$ that stops once the interval length is at most $\Delta$. Knowing that the number of iterations is $N = \lceil \log_2((b-a)/\Delta) \rceil$, at every iteration the algorithm makes a single call to `NoisyRC`$([a, \text{mid}], \mathcal{S})$, which computes a range count over the $n_{\text{cal}}$ calibration scores and adds Gaussian noise. With the scores stored in a simple array, this range count is a linear-time operation, so each iteration costs $\Theta(n_{\text{cal}})$, and the total running time is $\Theta(n_{\text{cal}}N) = \Theta(n_{\text{cal}} \log((b-a)/\Delta))$. In addition, apart from the stored scores, the algorithm keeps only a constant number of scalar variables (e.g., `left`, `right`, `mid`), so the auxiliary memory usage is $\mathcal{O}(1)$. Considering this computational complexity, since $\Delta$ is extremely small in practice (e.g., $10^{-10}$), the number of iterations is commonly around 30–40, making P-COQS computationally negligible compared to training $\hat{f}$ or its DP counterpart.

Having studied the rank error of the DP binary search subroutine and its computational complexity, we are now able to quantify to what extent the rank associated with the private quantile $q^{DP}$ from Algorithm 1 differs from the non-private rank in the calibration data. Based on this, we know that obtaining a $\rho$-zCDP quantile ensures that Algorithm 2 preserves the desired privacy level. Moreover, Proposition 1 allows us to explicitly define the DP noise error term $\tau^*$ with high probability (at least $1 - \beta$), which generally dominates $M(\Delta)$ since $\Delta$ is small. We will see that this information can be used and is indeed helpful when determining the coverage guarantees of P-COQS in Algorithm 2 which are stated in Theorem 1. For this purpose, we also define $s_{\text{test}} = s(X_{\text{test}}, Y, \hat{f})$.

**Theorem 1** (Coverage Guarantee). *Let $(s_1, \ldots, s_{n_{cal}}, s_{test})$ be an exchangeable sequence of non-conformity scores. With probability at least $1 - \beta$ (over the DP noise in Algorithm 1), the P-COQS prediction set $\mathcal{C}^{DP}(X_{test})$ satisfies:*

$$1 - \alpha - \frac{\tau}{n_{cal} + 1} \leq \mathbb{P}\big[Y_{test} \in \mathcal{C}^{DP}(X_{test})\big] \leq 1 - \alpha + \frac{\tau + 1}{n_{cal} + 1}.$$

*If ties are not randomized, the same guarantee holds with the lower bound replaced by $1 - \alpha - \frac{\tau + 1}{n_{cal} + 1}$.*

*Proof.* Recalling that $\tau = \tau^* + M(\Delta)$, where $\tau^*$ is the DP noise term from Proposition 1 and $M(\Delta)$ is the discretization term, we have that Algorithm 1 outputs a private quantile $q^{DP}$ with rank

error $\tau$, i.e. $s_{(r-\tau)} \leq q^{DP} \leq s_{(r+\tau)}$. By the exchangeability condition of the non-conformity scores, the rank of $s_{\text{test}} = s(X_{\text{test}}, Y, \hat{f})$ among $(s_1, \ldots, s_{n_{\text{cal}}}, s_{\text{test}})$ is uniformly distributed over $\{1, \ldots, n_{\text{cal}} + 1\}$. Thus, recalling that $r = \lceil (1 - \alpha)(n_{\text{cal}} + 1) \rceil$ and that

$$Y \in \mathcal{C}^{DP}(X_{\text{test}}) \iff s_{\text{test}} \leq q^{DP},$$

we consequently have:

$$\underbrace{\frac{r - \tau}{n_{\text{cal}} + 1}}_{L} \leq \mathbb{P}\left(s_{\text{test}} \leq q^{DP}\right) \leq \underbrace{\frac{r + \tau}{n_{\text{cal}} + 1}}_{U},$$

where:

$$L = \frac{\lceil (1 - \alpha)(n_{\text{cal}} + 1) \rceil - \tau}{n_{\text{cal}} + 1} \geq 1 - \alpha - \frac{\tau}{n_{\text{cal}} + 1} \quad \text{and}$$

$$U = \frac{\lceil (1 - \alpha)(n_{\text{cal}} + 1) \rceil + \tau}{n_{\text{cal}} + 1} \leq \frac{(1 - \alpha)(n_{\text{cal}} + 1) + 1 + \tau}{n_{\text{cal}} + 1} = 1 - \alpha + \frac{\tau + 1}{n_{\text{cal}} + 1}.$$

Therefore this allows us to conclude that

$$1 - \alpha - \frac{\tau}{n_{\text{cal}} + 1} \leq \mathbb{P}\left[Y_{\text{test}} \in \mathcal{C}^{DP}(X_{\text{test}})\right] \leq 1 - \alpha + \frac{\tau + 1}{n_{\text{cal}} + 1}.$$

$\square$

**Remark 6.** *Theorem 1 is a* marginal *coverage result: it holds under exchangeability of* $(s_1, \ldots, s_{n_{\text{cal}}}, s_{\text{test}})$ *and does not imply slice-wise (e.g., class-conditional or group-conditional) coverage, nor robustness under covariate shift. Extending P-COQS to Mondrian or other conditional conformal frameworks, or studying its behavior under distribution shift, is an interesting direction for future work but beyond the scope of this paper.*

Although Theorem 1 does not ensure the coverage probability $(1 - \alpha)$, as opposed to the method in Angelopoulos et al. (2022) that preserves this standard CP guarantee, it establishes an approximate coverage bound with an error term of order $\mathcal{O}(\tau/n_{\text{cal}})$. While this indeed implies an error margin for the actual coverage, it nevertheless also provides an upper bound to the coverage guarantee which allows to understand how conservative and how efficient the resulting prediction sets can be (see e.g. Lei et al., 2018; Li et al., 2025). More in detail, this error depends not only on the size of the calibration set but also on the width of the interval $[a, b]$ and the privacy level $\rho$ under zCDP. Specifically, as shown in Proposition 1, the coverage error increases with the length of the interval and decreases with the privacy budget. Therefore, achieving accurate coverage requires a sufficiently large calibration set when the parameters of the error bound $\tau$ are fixed. Nevertheless, all the experimental results in Section 5 indicate that P-COQS is able to achieve coverage comparable to the non-private conformal prediction method, despite this weaker theoretical guarantee.

**Remark 7.** *As highlighted by Huang et al. (2021), the quantile method adapted in Algorithm 1 is nearly optimal, implying that our results are suboptimal (polylogarithmic) in terms of CP guarantees. Nevertheless, using the bounds in Proposition 1, it delivers a practical way to determine the true differentially private coverage guarantee with high probability (i.e. $1 - \beta$) without having to inflate the coverage to mitigate the impact of privacy noise (which can negatively impact the informativeness of the CP sets). For example, suppose that we are dealing with a classification task where the*

*non-conformity scores lie in the interval* $[0, 1]$. *Then by fixing the other parameters* $\Delta$ *and* $\rho$ *(e.g.,* $\Delta = 10^{-10}$ *and* $\rho = 0.1$) *and assuming* $\min_{i,j} |s_i - s_j| < \Delta$ *(for all* $i, j \in \{1, \ldots, n_{cal}\}$) *and* $i \neq j$), *following Proposition 1 the true probability of DP coverage of P-COQS is at least* $1 - \alpha - 0.0183$ *and at most* $1 - \alpha + 0.0186$ *guaranteed with probability at least* $0.99$ *(if choosing* $\beta = 0.01$) *and calibration size* $n_{cal} = 3000$.

Following the above discussion, a straightforward adaptation of the significance level in Algorithm 2 can allow to guarantee the $1 - \alpha$ coverage (with probability at least $1 - \beta$) as stated in the corollary below.

**Corollary 1.** *Let*

$$\alpha^* = \max\Big\{0, \ \alpha - \frac{\tau}{n_{cal} + 1}\Big\}.$$

*Then, if we run Algorithm 2 with* $\alpha^*$ *in place of* $\alpha$, *with probability at least* $1 - \beta$ *we have that*

$$\mathbb{P}\left[Y_{test} \in \mathcal{C}^{DP}(X_{test})\right] \geq 1 - \alpha.$$

*If ties are not randomized, the same guarantee holds with* $\alpha^* = \max\{0, \alpha - \frac{\tau+1}{n_{cal}+1}\}$.

The proof of this corollary is a direct consequence of the bounds in Theorem 1 and is therefore omitted. As a result, similarly to Angelopoulos et al. (2022), we could also correct P-COQS to achieve the required coverage, although this would be achieved only with high probability $1 - \beta$ as opposed to Angelopoulos et al. (2022) where this coverage is fully guaranteed. More in detail though, the correction in Corollary 1 depends on the total rank-error bound $\tau = \tau^* + M(\Delta)$ where the discretization component $M(\Delta)$ is not directly observable since it depends on the calibration scores. Nevertheless, one can work with a deterministic or probabilistic upper bound on $M(\Delta)$ without compromising privacy. For instance, following Remark 2, if the score distribution admits a bounded density $L$ on $[a, b]$, then with high probability we have $\mathbb{E}[M(\Delta)] \leq L \Delta n_{\text{cal}}$, and a conservative choice such as $M(\Delta) \approx L \Delta n_{\text{cal}}$ can be used to adjust $\alpha^*$. Alternatively, if no distributional assumption is made, an auxiliary DP histogram over the $\Delta$-grid can provide a private estimate of $M(\Delta)$ at negligible additional privacy cost. This makes the resulting coverage slightly more conservative while still ensuring the nominal $1 - \alpha$ guarantee with high probability $1 - \beta$. However, in the following sections we do not make use of this correction since, as shown across all the experimental results, the empirical performance of P-COQS does not appear to be significantly affected by the original approximate guarantee stated in Theorem 1.

## 5 Experiments

In this section, we evaluate the proposed P-COQS under the same experimental setting as Angelopoulos et al. (2022)[1]. Additionally, we extend this evaluation with controlled simulation experiments. Specifically, we compare P-COQS with the method introduced by Angelopoulos et al. (2022) (henceforth referred to as EXPONQ) and to an additional benchmark represented by a DP histogram (CDF construction) approach to estimate a private quantile as an alternative subroutine for CP (Wasserman & Zhou, 2010): we refer to this approach as HISTLAP. Although the properties of the HISTLAP approach have not been formally investigated within a CP framework,

---

[1]The code to reproduce the experiments in this section and an R package implementation of P-COQS can be found at `https://github.com/SMAC-Group/pcoqs`

it is nevertheless a reasonable alternative to compare to the considered ExponQ and P-COQS approaches. This being said, we investigate scenarios where the CP approaches employ either non-private models ($\hat{f}$) or DP models ($\hat{f}^{\text{DP}}$). For differentially private models, we train Naïve Bayes and Random Forest models using IBM's `Diffprivlib` library in Python,[2] ensuring pure $\epsilon$-DP guarantees. For deep neural networks, we adopt the same methodology as Angelopoulos et al. (2022), utilizing the `Opacus` library to obtain $(\epsilon, \delta)$-DP models. For DP conformal prediction methods, we leverage the equivalence between $\epsilon$-DP and $(\epsilon, 0)$-zCDP established in Lemma 3.2 of Bun & Steinke (2016) where, among others, we have that $\epsilon$-DP implies ($\epsilon^2/2$)-zCDP. This allows us to analyze the CP approaches under either privacy framework while maintaining similar privacy guarantees. We specifically examine how these methods behave under varying privacy parameters ($\epsilon$) and different sample sizes. Consistent with the experimental design of Angelopoulos et al. (2022), we restrict our investigation to classification tasks. The non-conformity measure we used is given by the *hinge loss* function:

$$s(x, y, \hat{f}) = 1 - \hat{f}(x)_y,$$

where $\hat{f}(x)_y$ represents the predicted probability of the model for the true class $y$. We evaluate the performance of CP methods using the metrics of coverage, efficiency and informativeness. Therefore, if a CP approach performs well, we expect (i) the coverage to be close to $1 - \alpha$, (ii) the efficiency to be close to 1 from above, and (iii) the informativeness to be close to 1 from below. When comparing to the benchmark datasets considered in Angelopoulos et al. (2022), we employ their same experimental and parameters settings.

**Remark 8.** *When comparing P-COQS to the alternative benchmark HistLap (as well as to the existing ExponQ), a first aspect that must be underlined is that, in practice, they all require bounds on the non-conformity scores (i.e. $[a, b]$). However, as opposed to P-COQS, the HistLap approach (like ExponQ) requires the specification of the number of bins $B$ which must balance the discretization bias ($O(1/B)$) against the privacy noise term ($O(\sqrt{B}/(\epsilon_{CP} n_{cal}))$) and therefore needs to be optimized based on the sample size $n_{cal}$ and the privacy budget $\epsilon_{CP}$. In contrast, as pointed out in Remark 2, P-COQS only involves the parameter $\Delta$, whose effect on the privacy term is logarithmic ($\tau^* \propto \sqrt{\log((b-a)/\Delta)/\rho}$). Hence, $\Delta$ can be set very small (e.g., $10^{-10}$ or $1/n_{cal}$) without significantly affecting privacy noise, avoiding the need for tuning of an "optimal" discretization scale (see Proposition 1). Indeed, existing theoretical results for DP quantile estimation through privatized histograms (e.g. Wasserman & Zhou, 2010; Pillutla et al., 2022; Lalanne et al., 2023) provide explicit error bounds, typically of order $\tilde{O}((n_{cal}\varepsilon)^{-2/3})$ for the privacy-induced component of the error (after optimally choosing the number of bins $B$), together with discretization and sampling terms of orders $1/B$ and $n_{cal}^{-1/2}$, respectively. In contrast, the privacy contribution of Algorithm 1 based on the noisy range count (`NoisyRC`) decays as $\tilde{O}(1/(n_{cal}\sqrt{\rho}))$ under zCDP composition (with $\Delta \asymp c/n_{cal}$), up to logarithmic factors. Relating the two privacy measures through the standard conversion (Bun & Steinke, 2016), Algorithm 1 scales asymptotically faster in $n_{cal}$ than that of the DP histogram approach. Nevertheless, the total quantile error in both frameworks is ultimately dominated by the common sampling term $O(1/\sqrt{n_{cal}})$, so the advantage is most relevant in small samples and/or small privacy budgets.*

---

[2]DP implementations based on IBM's Diffprivlib: `https://diffprivlib.readthedocs.io/en/latest/modules/models.html#classification-models`

### 5.1 Simulated Data

We first consider a simple low-dimensional binary classification problem with eight features. The feature matrix for class 1 is sampled from a Gaussian $\mathcal{N}(\boldsymbol{\mu}_1, \boldsymbol{\Sigma}_1)$, with $\boldsymbol{\mu}_1 = [0.8, \ldots, 0.8]^T \in \mathbb{R}^8$ and $\boldsymbol{\Sigma}_1 = \mathbb{I} \cdot 7 \in \mathbb{R}^{8 \times 8}$, while class 2 follows $\mathcal{N}(\boldsymbol{\mu}_2, \boldsymbol{\Sigma}_2)$ with $\boldsymbol{\mu}_2 = [-1, \ldots, -1]$ and $\boldsymbol{\Sigma}_2 = \mathbb{I} \cdot 8$, where $\mathbb{I}$ denotes the identity matrix. This setting delivers a reasonable overlap between the two classes and implies a slightly non-linear decision boundary due to the small difference in the covariance matrices. The generated data is class-balanced, with 60% used for the training set, 24% for the calibration set, and 16% reserved for evaluating CP performance. As for the models, we consider the two classifiers mentioned earlier: Naïve Bayes (NB) and Random Forest (RF) (we will see applications with deep neural networks in Section 5.2). In particular, in this section we will focus on CP performance when employing the DP versions of these models (i.e. $\hat{f}^{DP}$), whereas the results using the non-DP models $\hat{f}$ can be found in the appendix. This being said, we evaluate coverage, efficiency, and informativeness with a fixed privacy budget $\epsilon_f = 2$ when privatizing the models and repeat experimental runs $H = 1,000$ times in each setting (unless otherwise specified). Moreover, when varying one of the parameters to study the sensitivity of the methods, the other parameters are fixed at: $\epsilon_{CP} = 1$ (privacy budget for CP, corresponding to ($\epsilon_{CP}^2/2$)-zCDP); $n = 10,000$ (total sample size to be split); $\alpha = 0.1$ (significance level for coverage). For the proposed P-COQS we use the default value for $\Delta$ in Algorithm 1 (i.e. $\Delta = 10^{-10}$) and in Appendix D we provide an ablation study with varying levels of $\Delta$ highlighting how the P-COQS performance appears to be stable for any reasonably small value of $\Delta$. For the HISTLAP approach, since there is no standard rule to choose the number of bins in a DP setting, we pre-evaluated a grid of values in $[20, 100]$ and chose to fix the number of bins to $B = 50$ which gave the best overall results and can be considered reasonable in practice since $n_{cal} = 2,400$ (implying that each bin contains on the order of tens of observations). For clarity of presentation, here we only discuss the results for the RF classifier, while we provide the same results for the NB classifier in Appendices A and B.

#### 5.1.1 Effect of CP Privacy Budget ($\epsilon_{CP}$)

We examine the sensitivity of P-COQS, EXPONQ and HISTLAP to variations in the CP privacy parameter $\epsilon_{CP}$, using a private RF model while maintaining a fixed sample size of $n = 10,000$. For the DP model (that achieves 79% accuracy), the results in Table 1 indicate that P-COQS and HISTLAP show significantly lower sensitivity to $\epsilon_{CP}$ compared to EXPONQ, while comparing favorably to it over the different evaluation metrics across the considered privacy budgets. In particular, for smaller values of $\epsilon_{CP}$, the P-COQS appears to perform better than HISTLAP as discussed earlier in Remark 8. Comparable results are observed for the private NB model (see Table 7 in Appendix A.1) where the P-COQS and HISTLAP have a more aligned behavior (while still outperforming EXPONQ), with P-COQS having a slight advantage for the smallest privacy budget. The same conclusions can be drawn when using the non-DP variants of the NB and RF models reported in Table 5 and Table 12 respectively where, as expected, efficiency and informativeness results are slightly better for all approaches.

#### 5.1.2 Effect of Sample Size ($n$)

We investigate the impact of sample size on the performance of the CP methods considered here. For each sample size, the sample is split as mentioned at the start of this section (i.e. 60% training, 24% calibration and 16% test). We recall that we discuss the results with the DP version of RF (see

| $\epsilon_{CP}$ | Coverage | | | Efficiency | | | Informativeness | | |
|---|---|---|---|---|---|---|---|---|---|
| | ExponQ | HistLap | P-COQS | ExponQ | HistLap | P-COQS | ExponQ | HistLap | P-COQS |
| 0.1 | $1.0000 \pm 0.0002$ | $0.9263 \pm 0.0195$ | $0.9008 \pm 0.0192$ | $1.9970 \pm 0.0087$ | $1.3582 \pm 0.0826$ | $1.2664 \pm 0.0669$ | $0.0030 \pm 0.0087$ | $0.6418 \pm 0.0826$ | $0.7336 \pm 0.0669$ |
| 0.5 | $0.9575 \pm 0.0272$ | $0.9046 \pm 0.0108$ | $0.8982 \pm 0.0111$ | $1.5621 \pm 0.2378$ | $1.2756 \pm 0.0427$ | $1.2550 \pm 0.0437$ | $0.4379 \pm 0.2378$ | $0.7244 \pm 0.0427$ | $0.7450 \pm 0.0437$ |
| 1.0 | $0.9242 \pm 0.0123$ | $0.9023 \pm 0.0105$ | $0.8977 \pm 0.0107$ | $1.3459 \pm 0.0555$ | $1.2679 \pm 0.0420$ | $1.2535 \pm 0.0438$ | $0.6541 \pm 0.0555$ | $0.7321 \pm 0.0420$ | $0.7465 \pm 0.0438$ |
| 1.5 | $0.9171 \pm 0.0105$ | $0.9016 \pm 0.0106$ | $0.8976 \pm 0.0106$ | $1.3183 \pm 0.0465$ | $1.2658 \pm 0.0421$ | $1.2533 \pm 0.0438$ | $0.6817 \pm 0.0465$ | $0.7342 \pm 0.0421$ | $0.7467 \pm 0.0438$ |
| 3.0 | $0.9096 \pm 0.0112$ | $0.9008 \pm 0.0104$ | $0.8976 \pm 0.0105$ | $1.2921 \pm 0.0471$ | $1.2633 \pm 0.0418$ | $1.2532 \pm 0.0441$ | $0.7079 \pm 0.0471$ | $0.7367 \pm 0.0418$ | $0.7468 \pm 0.0441$ |
| 5.0 | $0.9058 \pm 0.0121$ | $0.9004 \pm 0.0104$ | $0.8975 \pm 0.0104$ | $1.2801 \pm 0.0505$ | $1.2619 \pm 0.0417$ | $1.2529 \pm 0.0438$ | $0.7199 \pm 0.0505$ | $0.7381 \pm 0.0417$ | $0.7471 \pm 0.0438$ |
| 10.0 | $0.9034 \pm 0.0129$ | $0.9002 \pm 0.0104$ | $0.8976 \pm 0.0104$ | $1.2726 \pm 0.0526$ | $1.2613 \pm 0.0420$ | $1.2531 \pm 0.0440$ | $0.7274 \pm 0.0526$ | $0.7387 \pm 0.0420$ | $0.7469 \pm 0.0440$ |

Table 1: Average effect of CP privacy budget $\epsilon_{CP}$ on conformal prediction with DP Random Forest model. The numbers are metric averages over $1,000$ runs (per method and privacy budget) with variance bounds.

Appendix C.2 for the result of the non-DP model $\hat{f}$) and also recall that, as in other experiments, the privacy budget for the models is fixed at $\epsilon_f = 2$. The results in Table 2 show that in general P-COQS over-covers less than the DP alternatives in smaller sample sizes and hence converges more quickly towards the correct $1 - \alpha$ coverage, with consequent advantages over the DP alternatives in terms of efficiency and informativeness. In Appendices A.2 and B.2 the results for the NB classifier show that both the HistLap and P-COQS under-cover for the smallest sample size (i.e. $n = 100$) while ExponQ over-covers. Nevertheless, it can be observed that the P-COQS under-covering appears to be solved already starting from $n = 200$, as is the case for HistLap which however has a more inconsistent behavior as the sample size increases, under-covering in various occasions for moderate sample sizes.

| $n$ | Coverage | | | Efficiency | | | Informativeness | | |
|---|---|---|---|---|---|---|---|---|---|
| | ExponQ | HistLap | P-COQS | ExponQ | HistLap | P-COQS | ExponQ | HistLap | P-COQS |
| 100 | $0.9979 \pm 0.0142$ | $0.9979 \pm 0.0158$ | $0.9594 \pm 0.1040$ | $1.9977 \pm 0.0146$ | $1.9972 \pm 0.0236$ | $1.9404 \pm 0.1610$ | $0.0022 \pm 0.0146$ | $0.0027 \pm 0.0236$ | $0.0596 \pm 0.1610$ |
| 200 | $0.9993 \pm 0.0060$ | $0.9937 \pm 0.0232$ | $0.9634 \pm 0.0676$ | $1.9990 \pm 0.0078$ | $1.9895 \pm 0.0379$ | $1.9383 \pm 0.1120$ | $0.0010 \pm 0.0078$ | $0.0105 \pm 0.0379$ | $0.0617 \pm 0.1120$ |
| 500 | $0.9998 \pm 0.0019$ | $0.9626 \pm 0.0381$ | $0.9183 \pm 0.0550$ | $1.9983 \pm 0.0094$ | $1.8399 \pm 0.1317$ | $1.6949 \pm 0.1646$ | $0.0017 \pm 0.0094$ | $0.1601 \pm 0.1317$ | $0.3051 \pm 0.1646$ |
| 1000 | $0.9999 \pm 0.0011$ | $0.9307 \pm 0.0317$ | $0.9066 \pm 0.0331$ | $1.9977 \pm 0.0110$ | $1.6282 \pm 0.1188$ | $1.5448 \pm 0.1124$ | $0.0023 \pm 0.0110$ | $0.3718 \pm 0.1188$ | $0.4552 \pm 0.1124$ |
| 2000 | $0.9978 \pm 0.0065$ | $0.9165 \pm 0.0232$ | $0.9063 \pm 0.0228$ | $1.9678 \pm 0.0756$ | $1.4545 \pm 0.0913$ | $1.4190 \pm 0.0833$ | $0.0322 \pm 0.0756$ | $0.5455 \pm 0.0913$ | $0.5810 \pm 0.0833$ |
| 6000 | $0.9467 \pm 0.0258$ | $0.9041 \pm 0.0135$ | $0.9010 \pm 0.0134$ | $1.5064 \pm 0.1986$ | $1.3016 \pm 0.0578$ | $1.2915 \pm 0.0527$ | $0.4936 \pm 0.1986$ | $0.6984 \pm 0.0578$ | $0.7085 \pm 0.0527$ |
| 10000 | $0.9242 \pm 0.0123$ | $0.9022 \pm 0.0104$ | $0.8977 \pm 0.0107$ | $1.3459 \pm 0.0555$ | $1.2677 \pm 0.0417$ | $1.2535 \pm 0.0438$ | $0.6541 \pm 0.0555$ | $0.7323 \pm 0.0417$ | $0.7465 \pm 0.0438$ |

Table 2: Average effect of sample size on private conformal prediction with DP Random Forest model. The numbers are metric averages over $1,000$ runs (per method and sample size) and in parentheses is the corresponding variance of the metrics.

### 5.1.3 Effect of Significance Level ($\alpha$)

We examine the performance across varying $\alpha$ values (fixing $n = 10,000$, $\epsilon_f = 2$, and $\epsilon_{CP} = 1$). As seen in Table 3, P-COQS and HistLap appear to better target the required coverage level on average compared to ExponQ. With respect to the other metrics, the results indicate that P-COQS generally performs better than ExponQ and is comparable to the HistLap approach. Appendix B.3 provides similar conclusions the DP NB model.

### 5.1.4 Computational Efficiency

We compare the runtime of the considered methods (including the non-DP CP which we refer to as "Standard") under two settings which considered the non-DP and DP versions of each model (i.e. $\hat{f}$ and $\hat{f}^{DP}$ respectively). Confirming the model privacy budget to be $\epsilon_f = 2$, conformal privacy budget

| $\alpha$ | Coverage | | | Efficiency | | | Informativeness | | |
|---|---|---|---|---|---|---|---|---|---|
| | ExponQ | HistLap | P-COQS | ExponQ | HistLap | P-COQS | ExponQ | HistLap | P-COQS |
| 0.01 | 1.0000 (0.0002) | 0.9921 (0.0027) | 0.9904 (0.0038) | 1.9970 (0.0087) | 1.7839 (0.0353) | 1.7607 (0.0463) | 0.0030 (0.0087) | 0.2161 (0.0353) | 0.2393 (0.0463) |
| 0.05 | 0.9914 (0.0121) | 0.9523 (0.0076) | 0.9486 (0.0078) | 1.8486 (0.1751) | 1.4674 (0.0433) | 1.4487 (0.0438) | 0.1514 (0.1751) | 0.5326 (0.0433) | 0.5513 (0.0438) |
| 0.10 | 0.9242 (0.0123) | 0.9022 (0.0100) | 0.8977 (0.0107) | 1.3459 (0.0555) | 1.2677 (0.0408) | 1.2535 (0.0438) | 0.6541 (0.0555) | 0.7323 (0.0408) | 0.7465 (0.0438) |

Table 3: Average effect of $\alpha$ on private conformal prediction using DP Random Forest model. The numbers are metric averages over 1,000 runs (per method and significance level), with variances in parentheses.

$\epsilon_{CP} = 1$, total sample size $n = 10,000$ and significance level $\alpha = 0.1$, we record the average runtime over 1,000 runs. Table 4 highlights the considerable computational gain that P-COQS delivers with respect to ExponQ while remaining comparable (or often better) across all performance metrics considered above (similar conclusions hold for the NB model in Appendix B.4). It must be underlined that, as mentioned in Section 2, the increased computational time for ExponQ is mainly due to the optimization subroutine needed to choose the optimal number of bins and the quantile-inflation hyperparameter to achieve the smallest possible prediction set. This being said, we can observe that the HistLap approach is the most computationally efficient among the DP procedures while preserving similar performance to P-COQS: this is due to the fact that no optimal bin-width search procedure is run (as is the case for ExponQ) whereas P-COQS does not strictly require tuning of $\Delta$ and therefore loses some computational efficiency with respect to HistLap simply due to the binary-search. Even so, the computational difference can be considered almost negligible between P-COQS and HistLap since they have computational times in the order of milliseconds.

| | Coverage | Efficiency | Informativeness | Model Accuracy | Ave. time (secs) |
|---|---|---|---|---|---|
| $\hat{f}$ + Standard | $0.9025 \pm 0.0099$ | $1.2222 \pm 0.0224$ | $0.7778 \pm 0.0224$ | $0.8125 \pm 0.0093$ | $0.0001 \pm 0.0001$ |
| $\hat{f}$ + ExponQ | $0.9229 \pm 0.0120$ | $1.2939 \pm 0.0406$ | $0.7061 \pm 0.0406$ | $0.8125 \pm 0.0093$ | $0.7522 \pm 0.0075$ |
| $\hat{f}$ + HistLap | $0.9024 \pm 0.0101$ | $1.2220 \pm 0.0228$ | $0.7780 \pm 0.0228$ | $0.8125 \pm 0.0093$ | $0.0002 \pm 0.0001$ |
| $\hat{f}$ + P-COQS | $0.9025 \pm 0.0098$ | $1.2223 \pm 0.0224$ | $0.7777 \pm 0.0224$ | $0.8125 \pm 0.0093$ | $0.0076 \pm 0.0001$ |
| $\hat{f}^{DP}$ + Standard | $0.8965 \pm 0.0129$ | $1.2507 \pm 0.0573$ | $0.7493 \pm 0.0573$ | $0.7941 \pm 0.0173$ | $0.0001 \pm 0.0001$ |
| $\hat{f}^{DP}$ + ExponQ | $0.9242 \pm 0.0123$ | $1.3459 \pm 0.0555$ | $0.6541 \pm 0.0555$ | $0.7941 \pm 0.0173$ | $0.7570 \pm 0.0078$ |
| $\hat{f}^{DP}$ + HistLap | $0.9024 \pm 0.0103$ | $1.2682 \pm 0.0416$ | $0.7318 \pm 0.0416$ | $0.7941 \pm 0.0173$ | $0.0002 \pm 0.0001$ |
| $\hat{f}^{DP}$ + P-COQS | $0.8977 \pm 0.0106$ | $1.2533 \pm 0.0438$ | $0.7467 \pm 0.0438$ | $0.7941 \pm 0.0173$ | $0.0076 \pm 0.0001$ |

Table 4: Average computational time of Standard, ExponQ, HistLap and P-COQS combined with non-DP and DP versions of the RF classifier.

## 5.2 Datasets

In this section, we take the same settings as in Angelopoulos et al. (2022) and study the performance of P-COQS when applied to three benchmark datasets: CIFAR-10 (Krizhevsky et al., 2009), ImageNet (Deng et al., 2009) and CoronaHack (Pérez et al., 2020). We compare the performance of P-COQS with the non-DP CP approach (Standard) and with ExponQ proposed by Angelopoulos et al. (2022), and follow the same experimental setup as in the latter. Unless otherwise stated, the significance level is fixed at $\alpha = 0.1$, CP privacy budget at $\epsilon_{CP} = 1$ (corresponding to $(\epsilon_{CP}^2/2)$-

zCDP) and, when used, the DP models are trained to achieve $(\epsilon, \delta)$-DP using the `Opacus` library with ($\epsilon = 8$ and $\delta = 10^{-5}$).

### 5.2.1 CIFAR-10 Benchmark Analysis

We first evaluate P-COQS on the CIFAR-10 dataset, comparing its performance to EXPONQ under different privacy settings. Following Angelopoulos et al. (2022), we consider two scenarios for model training: (i) one where a non-DP model is used ($\hat{f}$) and (ii) one where a DP model is used ($\hat{f}^{DP}$). Both DP and non-DP models share the same convolutional neural network architecture. The DP model achieved an accuracy of 60% compared to 77% for the non-DP model. The evaluation is based on $1,000$ random splits of the CIFAR-10 validation set, each of size $n = 5,000$. Figure 1 displays the empirical coverage and prediction set sizes across these settings. Our findings suggest that P-COQS generally produces smaller prediction sets (Figure 1b), while achieving empirical coverage levels that are comparable to those of the standard (non-private) CP baseline (Figure 1a), except for the case where a DP model is used where P-COQS appears to slightly undercover with respect to the standard CP baseline. On the other hand, compared to P-COQS, by inflating the quantile EXPONQ guarantees the coverage level but tends to have larger prediction sets for this reason. In the latter case, for all methods it can be seen that prediction sets are larger when using a DP model (as expected).

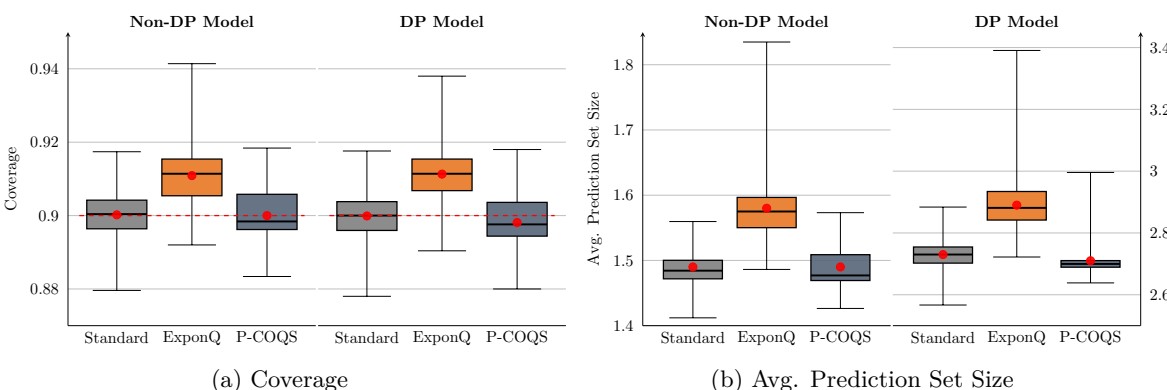

(a) Coverage

(b) Avg. Prediction Set Size

Figure 1: Results of Standard, EXPONQ and P-COQS methods applied to the CIFAR-10 dataset. (a) Boxplots with coverage distributions of the three methods under the non-DP model (left) and DP model (right) scenarios; (b) Boxplots with distributions of average set sizes of the three methods under the non-DP model (left) and DP model (right) scenarios. Red dots represent respective average metric.

### 5.2.2 Large-Scale Evaluation on ImageNet

We apply both considered private CP methods to the ImageNet dataset using a pre-trained (non-DP) ResNet-152 model. We evaluated the performance of P-COQS and EXPONQ under varying values of the conformal privacy budget $\epsilon_{CP}$. The evaluation uses $n = 30,000$ samples for calibration and $20,000$ samples for validation. For each budget value $\epsilon_{CP}$, performance metrics are computed over 100 random splits of the ImageNet validation set, following the procedure in Angelopoulos

et al. (2022). As shown in Figure 2, P-COQS appears to slightly under-cover but this does not appear to be significant based on the 95% confidence intervals (red vertical whiskers in the plots) and remains stable in performance across the different values of $\epsilon_{CP}$, whereas ExponQ appears to significantly over-cover for the small budgets (see Figure 2a). With this in mind, P-COQS also appears to be stable with respect to prediction set sizes while ExponQ approaches these sizes only as $\epsilon_{CP}$ increases (as highlighted also in the simulation results in Section 5.1).

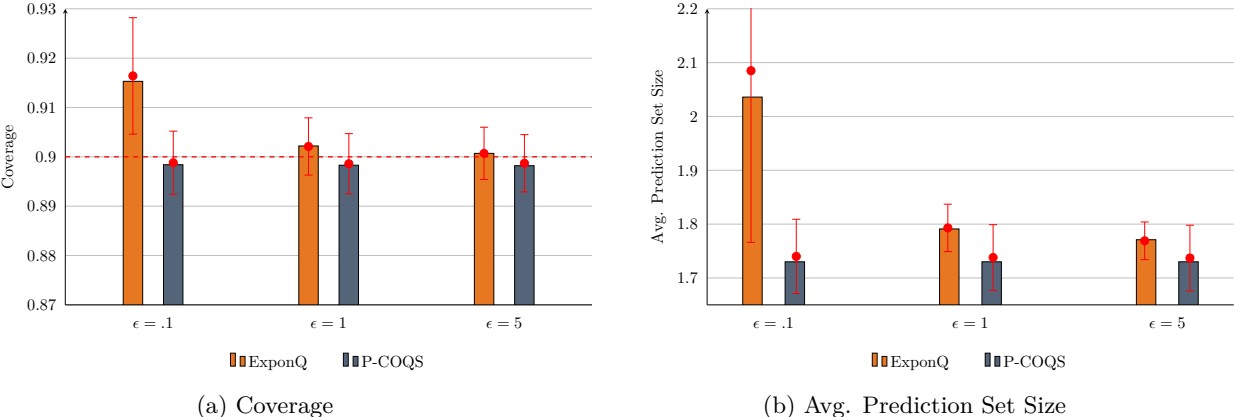

(a) Coverage
(b) Avg. Prediction Set Size

Figure 2: Results of ExponQ and P-COQS methods applied to the ImageNet dataset. (a) Median coverage (bar height), mean coverage (red dots) and 95% confidence intervals for coverage (red vertical whiskers) for each method under different privacy budgets $\epsilon$ (red-dashed horizontal line is the target coverage); (b) Average prediction set sizes (average set sizes in 100 splits): median (bar height), mean (red dots) and 95% confidence intervals (red vertical whiskers) for each method under different privacy budgets $\epsilon$.

### 5.2.3 Medical Imaging Analysis with CoronaHack Dataset

Finally, we evaluate both methods on the CoronaHack dataset, a publicly available chest X-ray dataset comprising 5,908 images categorized into three classes: normal, viral pneumonia (primarily COVID-19), and bacterial pneumonia. We fine-tuned the final layer of a ResNet-18 model over 14 epochs using 4,408 training samples, under both DP and non-DP settings. The DP model achieved 64% accuracy compared to 70% for the non-DP model. CP was calibrated on 1,000 samples and validated on 500 samples, across 1,000 random splits of the data. Figure 3 presents the results for different configurations of the DP and non-DP models. While ExponQ tends to produce overly conservative prediction sets, P-COQS appears to better target the nominal coverage level of 0.9 (see Figure 3a). Moreover, looking at the prediction set sizes, the distribution of the P-COQS appears to deliver roughly similar set sizes to the non-private CP approach since the proportions of sets of specific sizes (1, 2 or 3 on the x-axis) are very close to each other (see Figure 3b). Also in this case ExponQ produces larger sets (i.e. higher proportions for sets of size 2 or 3), where again the sizes tend to increase overall when using the DP model.

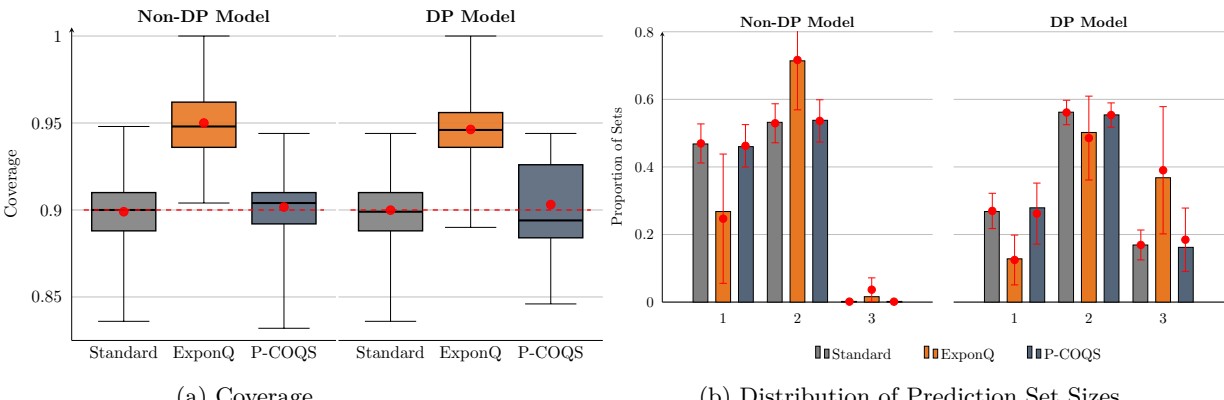

(a) Coverage  (b) Distribution of Prediction Set Sizes

Figure 3: Results of Standard, EXPONQ and P-COQS methods applied to the CoronaHack dataset. (a) Boxplots with coverage distributions of the three methods under the non-DP model (left) and DP model (right) scenarios; (b) Distribution of prediction set sizes where the y-axis represents the empirical proportion of sets of size 1, 2 or 3 (represented on the x-axis) for each method: median proportion (bar height), mean proportion (red dots) and 95% confidence intervals (red vertical whiskers) for each method under the non-DP model (left) and DP model (right) scenarios.

## 6  Discussion

The proposed P-COQS provides a computationally efficient DP alternative to existing privacy-preserving CP approaches. It does so by adapting an existing DP quantile method and trading off some theoretical guarantees on the lower bound for the coverage probability, while however guaranteeing quantifiable lower and upper error bounds for this probability. In this way, it is possible to determine (tight) error bounds which can then be used to assess the true minimum coverage level with high probability. Despite this theoretical trade-off, which however remains small (if not negligible) in common settings, the P-COQS shows an empirical performance over different metrics that is in line with standard non-private CP, indicating that it targets the correct coverage level in all settings while delivering more precise CP sets compared to the existing DP alternative (that we denoted as EXPONQ). The theoretical trade-off of the P-COQS can be observed mainly in settings with small sample sizes and/or privacy budgets in which the variability of P-COQS metrics is indeed larger compared to EXPONQ. However, the latter approach also makes a trade-off by generally over-covering with respect to the required coverage level in smaller samples and for small privacy budgets. As a consequence, the latter tends to deliver less efficient and informative prediction sets in these settings and appears to only approach those of P-COQS for larger privacy budgets and/or sample sizes. In general, considering these different trade-offs and without claiming the existence of a better approach, the proposed P-COQS appears to provide a valid privacy-preserving (and computationally efficient) CP approach also based on its generally good performance compared to EXPONQ in the different experimental settings considered. While both EXPONQ and P-COQS can be applied within different forms of CP (e.g. full CP), future work will aim to refine the theoretical coverage bounds of P-COQS in order to achieve the minimal $(1 - \alpha)$-coverage guarantees, without overly compromising the efficiency and informativeness of the

prediction sets nor the computational efficiency of the method, particularly when the calibration sample size and/or privacy budget are small.

## Acknowledgments

This work has been supported by the U.S. National Science Foundation under Grant No. SES-2150615. The authors would also like to greatly thank Aleksandra Slavković and Jordan Awan for their extremely helpful comments and feedback.

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

# A    Simulation Results With Non-DP Naïve Bayes Model

The parameter settings are as in the non-DP NB model and the model's accuracy is 83%.

## A.1    Effect of CP Privacy Budget ($\epsilon_{CP}$)

| $\epsilon_{CP}$ | Coverage | | | Efficiency | | | Informativeness | | |
|---|---|---|---|---|---|---|---|---|---|
| | ExponQ | HistLap | P-COQS | ExponQ | HistLap | P-COQS | ExponQ | HistLap | P-COQS |
| 0.1 | $0.9999 \pm 0.0004$ | $0.9066 \pm 0.0199$ | $0.9005 \pm 0.0190$ | $1.9678 \pm 0.0289$ | $1.2005 \pm 0.0559$ | $1.1813 \pm 0.0529$ | $0.0322 \pm 0.0289$ | $0.7995 \pm 0.0559$ | $0.8187 \pm 0.0529$ |
| 0.5 | $0.9444 \pm 0.0174$ | $0.9001 \pm 0.0108$ | $0.9005 \pm 0.0103$ | $1.3465 \pm 0.0972$ | $1.1778 \pm 0.0243$ | $1.1787 \pm 0.0220$ | $0.6535 \pm 0.0972$ | $0.8222 \pm 0.0243$ | $0.8213 \pm 0.0220$ |
| 1.0 | $0.9227 \pm 0.0116$ | $0.9002 \pm 0.0100$ | $0.9005 \pm 0.0101$ | $1.2509 \pm 0.0360$ | $1.1779 \pm 0.0205$ | $1.1788 \pm 0.0206$ | $0.7491 \pm 0.0360$ | $0.8221 \pm 0.0205$ | $0.8212 \pm 0.0206$ |
| 1.5 | $0.9154 \pm 0.0103$ | $0.9002 \pm 0.0099$ | $0.9006 \pm 0.0099$ | $1.2254 \pm 0.0267$ | $1.1776 \pm 0.0202$ | $1.1789 \pm 0.0201$ | $0.7746 \pm 0.0267$ | $0.8224 \pm 0.0202$ | $0.8211 \pm 0.0201$ |
| 3.0 | $0.9081 \pm 0.0097$ | $0.9002 \pm 0.0098$ | $0.9006 \pm 0.0099$ | $1.2019 \pm 0.0213$ | $1.1777 \pm 0.0197$ | $1.1789 \pm 0.0200$ | $0.7981 \pm 0.0213$ | $0.8223 \pm 0.0197$ | $0.8211 \pm 0.0200$ |
| 5.0 | $0.9051 \pm 0.0097$ | $0.9002 \pm 0.0098$ | $0.9006 \pm 0.0099$ | $1.1925 \pm 0.0203$ | $1.1777 \pm 0.0196$ | $1.1788 \pm 0.0200$ | $0.8075 \pm 0.0203$ | $0.8223 \pm 0.0196$ | $0.8212 \pm 0.0200$ |
| 10.0 | $0.9029 \pm 0.0097$ | $0.9002 \pm 0.0098$ | $0.9006 \pm 0.0099$ | $1.1857 \pm 0.0198$ | $1.1777 \pm 0.0197$ | $1.1788 \pm 0.0201$ | $0.8143 \pm 0.0198$ | $0.8223 \pm 0.0197$ | $0.8212 \pm 0.0201$ |

Table 5: Average effect of CP privacy budget $\epsilon_{CP}$ on conformal prediction with non-DP Naïve Bayes model. The numbers are metric averages over 1000 runs (per method and privacy budget) and the corresponding variance bounds of the metrics.

## A.2    Effect of Sample Size ($n$)

| Samples | Coverage | | | Efficiency | | | Informativeness | | |
|---|---|---|---|---|---|---|---|---|---|
| | ExponQ | HistLap | P-COQS | ExponQ | HistLap | P-COQS | ExponQ | HistLap | P-COQS |
| 100 | $0.9794 \pm 0.0434$ | $0.9332 \pm 0.0723$ | $0.8664 \pm 0.1230$ | $1.7300 \pm 0.2179$ | $1.4304 \pm 0.1911$ | $1.2774 \pm 0.2738$ | $0.2700 \pm 0.2179$ | $0.5696 \pm 0.1911$ | $0.7226 \pm 0.2738$ |
| 200 | $0.9905 \pm 0.0207$ | $0.9367 \pm 0.0507$ | $0.8985 \pm 0.0898$ | $1.7901 \pm 0.1638$ | $1.3753 \pm 0.1469$ | $1.2892 \pm 0.2349$ | $0.2099 \pm 0.1638$ | $0.6247 \pm 0.1469$ | $0.7108 \pm 0.2349$ |
| 500 | $0.9967 \pm 0.0078$ | $0.9224 \pm 0.0383$ | $0.9036 \pm 0.0539$ | $1.8566 \pm 0.1158$ | $1.2780 \pm 0.1027$ | $1.2292 \pm 0.1352$ | $0.1434 \pm 0.1158$ | $0.7220 \pm 0.1027$ | $0.7708 \pm 0.1352$ |
| 1000 | $0.9984 \pm 0.0042$ | $0.9075 \pm 0.0325$ | $0.9026 \pm 0.0350$ | $1.8951 \pm 0.0876$ | $1.2146 \pm 0.0780$ | $1.2006 \pm 0.0806$ | $0.1049 \pm 0.0876$ | $0.7854 \pm 0.0780$ | $0.7994 \pm 0.0806$ |
| 2000 | $0.9836 \pm 0.0129$ | $0.9041 \pm 0.0230$ | $0.9031 \pm 0.0227$ | $1.6272 \pm 0.1483$ | $1.1932 \pm 0.0526$ | $1.1905 \pm 0.0520$ | $0.3728 \pm 0.1483$ | $0.8068 \pm 0.0526$ | $0.8095 \pm 0.0520$ |
| 6000 | $0.9382 \pm 0.0169$ | $0.9002 \pm 0.0128$ | $0.9009 \pm 0.0127$ | $1.3149 \pm 0.0795$ | $1.1782 \pm 0.0279$ | $1.1801 \pm 0.0282$ | $0.6851 \pm 0.0795$ | $0.8218 \pm 0.0279$ | $0.8199 \pm 0.0282$ |
| 10000 | $0.9227 \pm 0.0116$ | $0.9002 \pm 0.0101$ | $0.9005 \pm 0.0101$ | $1.2509 \pm 0.0360$ | $1.1778 \pm 0.0207$ | $1.1788 \pm 0.0206$ | $0.7491 \pm 0.0360$ | $0.8222 \pm 0.0207$ | $0.8212 \pm 0.0206$ |

Table 6: Average effect of sample size on DP CP ( $\epsilon_{CP} = 1$) with non-DP Naïve Bayes model. The numbers are metric averages over 1000 runs (per method and sample size) and the corresponding variance bounds of the metrics.

# B Simulation Results With DP Naïve Bayes Model

The parameter settings are as in the DP RF model and the model's accuracy is 75%.

## B.1 Effect of CP Privacy Budget ($\epsilon_{CP}$)

| $\epsilon_{CP}$ | Coverage | | | Efficiency | | | Informativeness | | |
|---|---|---|---|---|---|---|---|---|---|
| | ExponQ | HistLap | P-COQS | ExponQ | HistLap | P-COQS | ExponQ | HistLap | P-COQS |
| 0.1 | $1.0000 \pm 0.0002$ | $0.8985 \pm 0.0162$ | $0.8990 \pm 0.0197$ | $1.9995 \pm 0.0030$ | $1.3954 \pm 0.0555$ | $1.3981 \pm 0.0677$ | $0.0005 \pm 0.0030$ | $0.6046 \pm 0.0555$ | $0.6019 \pm 0.0677$ |
| 0.5 | $0.9402 \pm 0.0141$ | $0.8995 \pm 0.0101$ | $0.8999 \pm 0.0104$ | $1.5692 \pm 0.0694$ | $1.3969 \pm 0.0336$ | $1.3988 \pm 0.0341$ | $0.4308 \pm 0.0694$ | $0.6031 \pm 0.0336$ | $0.6012 \pm 0.0341$ |
| 1.0 | $0.9217 \pm 0.0116$ | $0.8995 \pm 0.0098$ | $0.8999 \pm 0.0100$ | $1.4841 \pm 0.0457$ | $1.3970 \pm 0.0316$ | $1.3985 \pm 0.0326$ | $0.5159 \pm 0.0457$ | $0.6030 \pm 0.0316$ | $0.6015 \pm 0.0326$ |
| 1.5 | $0.9148 \pm 0.0104$ | $0.8995 \pm 0.0097$ | $0.8999 \pm 0.0098$ | $1.4553 \pm 0.0385$ | $1.3970 \pm 0.0316$ | $1.3985 \pm 0.0321$ | $0.5447 \pm 0.0385$ | $0.6030 \pm 0.0316$ | $0.6015 \pm 0.0321$ |
| 3.0 | $0.9076 \pm 0.0098$ | $0.8996 \pm 0.0096$ | $0.8999 \pm 0.0097$ | $1.4272 \pm 0.0337$ | $1.3972 \pm 0.0314$ | $1.3985 \pm 0.0320$ | $0.5728 \pm 0.0337$ | $0.6028 \pm 0.0314$ | $0.6015 \pm 0.0320$ |
| 5.0 | $0.9046 \pm 0.0097$ | $0.8995 \pm 0.0096$ | $0.8999 \pm 0.0097$ | $1.4160 \pm 0.0325$ | $1.3971 \pm 0.0315$ | $1.3986 \pm 0.0320$ | $0.5840 \pm 0.0325$ | $0.6029 \pm 0.0315$ | $0.6014 \pm 0.0320$ |
| 10.0 | $0.9024 \pm 0.0096$ | $0.8995 \pm 0.0096$ | $0.9000 \pm 0.0097$ | $1.4073 \pm 0.0321$ | $1.3971 \pm 0.0315$ | $1.3987 \pm 0.0321$ | $0.5927 \pm 0.0321$ | $0.6029 \pm 0.0315$ | $0.6013 \pm 0.0321$ |

Table 7: Average effect of CP privacy budget $\epsilon_{CP}$ on conformal prediction with DP Naïve Bayes model and $\epsilon_f = 2$. The numbers are metric averages over 1000 runs (per method and privacy budget) and the corresponding variance bounds of the metrics.

## B.2 Effect of Sample Size ($n$)

| Samples | Coverage | | | Efficiency | | | Informativeness | | |
|---|---|---|---|---|---|---|---|---|---|
| | ExponQ | HistLap | P-COQS | ExponQ | HistLap | P-COQS | ExponQ | HistLap | P-COQS |
| 100 | $0.9791 \pm 0.0447$ | $0.8037 \pm 0.1297$ | $0.7984 \pm 0.1651$ | $1.9266 \pm 0.1180$ | $1.4537 \pm 0.2054$ | $1.4657 \pm 0.3283$ | $0.0734 \pm 0.1180$ | $0.5463 \pm 0.2054$ | $0.5343 \pm 0.3283$ |
| 200 | $0.9954 \pm 0.0148$ | $0.9443 \pm 0.0650$ | $0.9235 \pm 0.0830$ | $1.9896 \pm 0.0287$ | $1.8542 \pm 0.1435$ | $1.7948 \pm 0.1916$ | $0.0104 \pm 0.0287$ | $0.1458 \pm 0.1435$ | $0.2052 \pm 0.1916$ |
| 500 | $0.9969 \pm 0.0074$ | $0.9038 \pm 0.0458$ | $0.9038 \pm 0.0517$ | $1.9830 \pm 0.0274$ | $1.6111 \pm 0.1082$ | $1.6175 \pm 0.1356$ | $0.0170 \pm 0.0274$ | $0.3889 \pm 0.1082$ | $0.3825 \pm 0.1356$ |
| 1000 | $0.9994 \pm 0.0034$ | $0.8695 \pm 0.0628$ | $0.9261 \pm 0.0556$ | $1.9963 \pm 0.0147$ | $1.5900 \pm 0.1060$ | $1.7337 \pm 0.1874$ | $0.0037 \pm 0.0147$ | $0.4100 \pm 0.1060$ | $0.2663 \pm 0.1874$ |
| 2000 | $0.9761 \pm 0.0130$ | $0.8987 \pm 0.0219$ | $0.9012 \pm 0.0227$ | $1.8469 \pm 0.0577$ | $1.5223 \pm 0.0638$ | $1.5313 \pm 0.0669$ | $0.1531 \pm 0.0577$ | $0.4777 \pm 0.0638$ | $0.4687 \pm 0.0669$ |
| 6000 | $0.9340 \pm 0.0178$ | $0.8873 \pm 0.0211$ | $0.9190 \pm 0.0417$ | $1.6877 \pm 0.0859$ | $1.5088 \pm 0.0341$ | $1.6233 \pm 0.1906$ | $0.3123 \pm 0.0859$ | $0.4912 \pm 0.0341$ | $0.3767 \pm 0.1906$ |
| 10000 | $0.9217 \pm 0.0116$ | $0.8994 \pm 0.0098$ | $0.8999 \pm 0.0100$ | $1.4841 \pm 0.0457$ | $1.3968 \pm 0.0321$ | $1.3985 \pm 0.0326$ | $0.5159 \pm 0.0457$ | $0.6032 \pm 0.0321$ | $0.6015 \pm 0.0326$ |

Table 8: Average effect of sample size on DP CP ( $\epsilon_{CP} = 1$) with DP Naïve Bayes model ($\epsilon_f = 2$). The numbers are metric averages over 1000 runs (per method and sample size) and the corresponding variance bounds of the metrics.

## B.3 Effect of Significance Level ($\alpha$)

| $\alpha$ | Coverage | | | Efficiency | | | Informativeness | | |
|---|---|---|---|---|---|---|---|---|---|
| | ExponQ | HistLap | P-COQS | ExponQ | HistLap | P-COQS | ExponQ | HistLap | P-COQS |
| 0.01 | 1.0000 (0.0002) | 0.9822 (0.0047) | 0.9985 (0.0050) | 1.9995 (0.0030) | 1.8228 (0.0136) | 1.9824 (0.0562) | 0.0005 (0.0030) | 0.1772 (0.0136) | 0.0176 (0.0562) |
| 0.05 | 0.9732 (0.0088) | 0.9496 (0.0072) | 0.9500 (0.0078) | 1.7579 (0.0604) | 1.6129 (0.0337) | 1.6153 (0.0380) | 0.2421 (0.0604) | 0.3871 (0.0337) | 0.3847 (0.0380) |
| 0.10 | 0.9217 (0.0116) | 0.9009 (0.0096) | 0.8999 (0.0100) | 1.4841 (0.0457) | 1.4022 (0.0314) | 1.3985 (0.0326) | 0.5159 (0.0457) | 0.5978 (0.0314) | 0.6015 (0.0326) |

Table 9: Average effect of $\alpha$ on DP CP ($\epsilon_{CP} = 1$) with DP Naïve Bayes model ($\epsilon_f = 2$). The numbers are metric averages over 1000 runs (per method and $\alpha$ value) and the corresponding variance bounds of the metrics.

### B.4   Average CP computational time with DP and non-DP models

|  | Coverage | Efficiency | Informativeness | Model Accuracy | Ave. time (secs) |
|---|---|---|---|---|---|
| $\hat{f}$ + Standard | $0.9025 \pm 0.0099$ | $1.2222 \pm 0.0224$ | $0.7778 \pm 0.0224$ | $0.8125 \pm 0.0093$ | $0.0001 \pm 0.0001$ |
| $\hat{f}$ + ExponQ | $0.9229 \pm 0.0120$ | $1.2939 \pm 0.0406$ | $0.7061 \pm 0.0406$ | $0.8125 \pm 0.0093$ | $0.7522 \pm 0.0075$ |
| $\hat{f}$ + HistLap | $0.9024 \pm 0.0101$ | $1.2220 \pm 0.0228$ | $0.7780 \pm 0.0228$ | $0.8125 \pm 0.0093$ | $0.0002 \pm 0.0001$ |
| $\hat{f}$ + P-COQS | $0.9025 \pm 0.0098$ | $1.2223 \pm 0.0224$ | $0.7777 \pm 0.0224$ | $0.8125 \pm 0.0093$ | $0.0076 \pm 0.0001$ |
| $\hat{f}^{DP}$ + Standard | $0.8965 \pm 0.0129$ | $1.2507 \pm 0.0573$ | $0.7493 \pm 0.0573$ | $0.7941 \pm 0.0173$ | $0.0001 \pm 0.0001$ |
| $\hat{f}^{DP}$ + ExponQ | $0.9242 \pm 0.0123$ | $1.3459 \pm 0.0555$ | $0.6541 \pm 0.0555$ | $0.7941 \pm 0.0173$ | $0.7570 \pm 0.0078$ |
| $\hat{f}^{DP}$ + HistLap | $0.9024 \pm 0.0103$ | $1.2682 \pm 0.0416$ | $0.7318 \pm 0.0416$ | $0.7941 \pm 0.0173$ | $0.0002 \pm 0.0001$ |
| $\hat{f}^{DP}$ + P-COQS | $0.8977 \pm 0.0106$ | $1.2533 \pm 0.0438$ | $0.7467 \pm 0.0438$ | $0.7941 \pm 0.0173$ | $0.0076 \pm 0.0001$ |

Table 10: Average computational time of Standard, ExponQ, HistLap and P-COQS and combined with non-DP and DP versions of the Naïve Bayes classifier. The numbers are metric averages over 1000 runs (per setting) and in parentheses is the corresponding variance of the metrics.

### B.5   Effect of Model Privacy Budget ($\epsilon_f$)

| $\epsilon_f$ | Coverage | | | Efficiency | | | Informativeness | | |
|---|---|---|---|---|---|---|---|---|---|
| | ExponQ | HistLap | P-COQS | ExponQ | HistLap | P-COQS | ExponQ | HistLap | P-COQS |
| 0.1 | 0.9218 (0.0115) | 0.8991 (0.0098) | 0.8995 (0.0102) | 1.6580 (0.0335) | 1.5895 (0.0226) | 1.5907 (0.0238) | 0.3420 (0.0335) | 0.4105 (0.0226) | 0.4093 (0.0238) |
| 0.5 | 0.9222 (0.0110) | 0.8995 (0.0094) | 0.8999 (0.0096) | 1.5361 (0.0389) | 1.4516 (0.0259) | 1.4530 (0.0264) | 0.4639 (0.0389) | 0.5484 (0.0259) | 0.5470 (0.0264) |
| 1 | 0.9405 (0.0366) | 0.8394 (0.0554) | 0.9812 (0.0409) | 1.7647 (0.1443) | 1.4575 (0.0753) | 1.9208 (0.1717) | 0.2353 (0.1443) | 0.5425 (0.0753) | 0.0792 (0.1717) |
| 2 | 0.9217 (0.0116) | 0.8995 (0.0098) | 0.8999 (0.0100) | 1.4841 (0.0457) | 1.3971 (0.0317) | 1.3985 (0.0326) | 0.5159 (0.0457) | 0.6029 (0.0317) | 0.6015 (0.0326) |
| 5 | 0.9224 (0.0117) | 0.8997 (0.0098) | 0.9000 (0.0099) | 1.2784 (0.0372) | 1.2026 (0.0217) | 1.2035 (0.0218) | 0.7216 (0.0372) | 0.7974 (0.0217) | 0.7965 (0.0218) |
| 10 | 0.9225 (0.0116) | 0.8999 (0.0099) | 0.9003 (0.0098) | 1.2570 (0.0361) | 1.1831 (0.0207) | 1.1841 (0.0205) | 0.7430 (0.0361) | 0.8169 (0.0207) | 0.8159 (0.0205) |

Table 11: Average effect of model privacy budget $\epsilon_f$ on DP CP ($\epsilon_{CP} = 1$) with DP Naïve Bayes model. The numbers are metric averages over 1000 runs (per method and privacy budget) and the corresponding variance bounds of the metrics.

# C   Simulation Results With Non-DP Random Forest Model

The parameter settings are as described in Section 5.1 and the accuracy of the model is 81%.

## C.1   Effect of Privacy Budget ($\epsilon_{CP}$)

| $\epsilon_{CP}$ | Coverage | | | Efficiency | | | Informativeness | | |
|---|---|---|---|---|---|---|---|---|---|
| | ExponQ | HistLap | P-COQS | ExponQ | HistLap | P-COQS | ExponQ | HistLap | P-COQS |
| 0.1 | $0.9998 \pm 0.0004$ | $0.9126 \pm 0.0191$ | $0.9027 \pm 0.0195$ | $1.9886 \pm 0.0123$ | $1.2599 \pm 0.0625$ | $1.2263 \pm 0.0593$ | $0.0114 \pm 0.0123$ | $0.7401 \pm 0.0625$ | $0.7737 \pm 0.0593$ |
| 0.5 | $0.9464 \pm 0.0189$ | $0.9028 \pm 0.0107$ | $0.9023 \pm 0.0105$ | $1.4095 \pm 0.1217$ | $1.2235 \pm 0.0257$ | $1.2219 \pm 0.0245$ | $0.5905 \pm 0.1217$ | $0.7765 \pm 0.0257$ | $0.7781 \pm 0.0245$ |
| 1.0 | $0.9229 \pm 0.0120$ | $0.9026 \pm 0.0101$ | $0.9024 \pm 0.0099$ | $1.2939 \pm 0.0406$ | $1.2225 \pm 0.0231$ | $1.2219 \pm 0.0225$ | $0.7061 \pm 0.0406$ | $0.7775 \pm 0.0231$ | $0.7781 \pm 0.0225$ |
| 1.5 | $0.9155 \pm 0.0106$ | $0.9023 \pm 0.0100$ | $0.9025 \pm 0.0098$ | $1.2659 \pm 0.0295$ | $1.2218 \pm 0.0227$ | $1.2223 \pm 0.0224$ | $0.7341 \pm 0.0295$ | $0.7782 \pm 0.0227$ | $0.7777 \pm 0.0224$ |
| 3.0 | $0.9083 \pm 0.0099$ | $0.9022 \pm 0.0100$ | $0.9026 \pm 0.0098$ | $1.2411 \pm 0.0236$ | $1.2214 \pm 0.0224$ | $1.2225 \pm 0.0222$ | $0.7589 \pm 0.0236$ | $0.7786 \pm 0.0224$ | $0.7775 \pm 0.0222$ |
| 5.0 | $0.9057 \pm 0.0098$ | $0.9023 \pm 0.0099$ | $0.9026 \pm 0.0098$ | $1.2324 \pm 0.0226$ | $1.2216 \pm 0.0224$ | $1.2227 \pm 0.0223$ | $0.7676 \pm 0.0226$ | $0.7784 \pm 0.0224$ | $0.7773 \pm 0.0223$ |
| 10.0 | $0.9040 \pm 0.0098$ | $0.9023 \pm 0.0099$ | $0.9026 \pm 0.0099$ | $1.2270 \pm 0.0222$ | $1.2216 \pm 0.0222$ | $1.2227 \pm 0.0224$ | $0.7730 \pm 0.0222$ | $0.7784 \pm 0.0222$ | $0.7773 \pm 0.0224$ |

Table 12: Average effect of CP privacy budget $\epsilon_{CP}$ on conformal prediction with non-DP Random Forest model. The numbers are metric averages over 1000 runs (per method and privacy budget) and in parentheses is the corresponding variance of the metrics.

## C.2   Effect of Sample Size ($n$)

| $n$ | Coverage | | | Efficiency | | | Informativeness | | |
|---|---|---|---|---|---|---|---|---|---|
| | ExponQ | HistLap | P-COQS | ExponQ | HistLap | P-COQS | ExponQ | HistLap | P-COQS |
| 100 | $0.9892 \pm 0.0326$ | $0.9777 \pm 0.0451$ | $0.8905 \pm 0.1327$ | $1.8658 \pm 0.1716$ | $1.7618 \pm 0.1932$ | $1.4777 \pm 0.3624$ | $0.1342 \pm 0.1716$ | $0.2382 \pm 0.1932$ | $0.5222 \pm 0.3624$ |
| 200 | $0.9931 \pm 0.0183$ | $0.9655 \pm 0.0427$ | $0.9129 \pm 0.0905$ | $1.8925 \pm 0.1308$ | $1.6288 \pm 0.1828$ | $1.4559 \pm 0.2947$ | $0.1075 \pm 0.1308$ | $0.3712 \pm 0.1828$ | $0.5441 \pm 0.2947$ |
| 500 | $0.9973 \pm 0.0076$ | $0.9370 \pm 0.0397$ | $0.9094 \pm 0.0537$ | $1.9274 \pm 0.0872$ | $1.4248 \pm 0.1340$ | $1.3273 \pm 0.1627$ | $0.0726 \pm 0.0872$ | $0.5752 \pm 0.1340$ | $0.6727 \pm 0.1627$ |
| 1000 | $0.9987 \pm 0.0036$ | $0.9175 \pm 0.0296$ | $0.9058 \pm 0.0340$ | $1.9520 \pm 0.0577$ | $1.3131 \pm 0.0859$ | $1.2737 \pm 0.0929$ | $0.0480 \pm 0.0577$ | $0.6869 \pm 0.0859$ | $0.7263 \pm 0.0929$ |
| 2000 | $0.9865 \pm 0.0126$ | $0.9083 \pm 0.0228$ | $0.9055 \pm 0.0233$ | $1.7402 \pm 0.1492$ | $1.2600 \pm 0.0595$ | $1.2512 \pm 0.0607$ | $0.2598 \pm 0.1492$ | $0.7400 \pm 0.0595$ | $0.7488 \pm 0.0607$ |
| 6000 | $0.9393 \pm 0.0181$ | $0.9028 \pm 0.0127$ | $0.9028 \pm 0.0125$ | $1.3745 \pm 0.0986$ | $1.2274 \pm 0.0307$ | $1.2275 \pm 0.0300$ | $0.6255 \pm 0.0986$ | $0.7726 \pm 0.0307$ | $0.7725 \pm 0.0300$ |
| 10000 | $0.9229 \pm 0.0120$ | $0.9025 \pm 0.0101$ | $0.9024 \pm 0.0099$ | $1.2939 \pm 0.0406$ | $1.2223 \pm 0.0235$ | $1.2219 \pm 0.0225$ | $0.7061 \pm 0.0406$ | $0.7777 \pm 0.0235$ | $0.7781 \pm 0.0225$ |

Table 13: Average effect of sample size on DP CP ($\epsilon_{CP} = 1$) with non-DP Random Forest model. The numbers are metric averages over 1000 runs (per method and sample size) and in parentheses is the corresponding variance of the metrics.

## D   Ablation Study on $\Delta$

In this appendix we provide a brief ablation study to understand the sensitivity of Algorithm 1 to the choice of $\Delta$. Using a range for $\Delta$ between $10^{-20}$ to $10^{-1}$ and the same simulation settings as in Section 5.1, the results are presented in Table 14 below.

|  | $\Delta$ | Coverage | Efficiency | Informativeness | Accuracy |
|---|---|---|---|---|---|
| Naive Bayes | $10 \times 10^{-20}$ | 0.9219 (0.0109) | 1.4856 (0.0534) | 0.5144 (0.0534) | 0.7479 (0.0125) |
|  | $10 \times 10^{-15}$ | 0.9219 (0.0109) | 1.4856 (0.0534) | 0.5144 (0.0534) | 0.7479 (0.0125) |
|  | $10 \times 10^{-10}$ | 0.9219 (0.0109) | 1.4856 (0.0534) | 0.5144 (0.0534) | 0.7479 (0.0125) |
|  | $10 \times 10^{-5}$ | 0.9219 (0.0109) | 1.4856 (0.0534) | 0.5144 (0.0534) | 0.7479 (0.0125) |
|  | $10 \times 10^{-1}$ | 1.0000 (0.0000) | 2.0000 (0.0000) | 0.0000 (0.0000) | 0.7479 (0.0125) |
| Random Forest | $10 \times 10^{-20}$ | 0.8979 (0.0105) | 1.2542 (0.0436) | 0.7458 (0.0436) | 0.7941 (0.0173) |
|  | $10 \times 10^{-15}$ | 0.8979 (0.0105) | 1.2542 (0.0436) | 0.7458 (0.0436) | 0.7941 (0.0173) |
|  | $10 \times 10^{-10}$ | 0.8979 (0.0105) | 1.2542 (0.0436) | 0.7458 (0.0436) | 0.7941 (0.0173) |
|  | $10 \times 10^{-5}$ | 0.8979 (0.0105) | 1.2542 (0.0436) | 0.7458 (0.0436) | 0.7941 (0.0173) |
|  | $10 \times 10^{-1}$ | 0.8983 (0.0106) | 1.2554 (0.0447) | 0.7446 (0.0447) | 0.7941 (0.0173) |

Table 14: Average effect of parameter $\Delta$ on P-COQS with DP Naive Bayes and Random Forest models ($\epsilon_f = 2$ and $\epsilon_{CP} = 1$). The numbers are metric averages over 1000 runs and in parentheses are the corresponding variance of the metrics.

We can observe that the performance of P-COQS remains unchanged for all reasonably small values of $\Delta$ while it changes only for the value $\Delta = 10^{-1}$: this is not surprising since the range for the non-conformity scores is $[0, 1]$, meaning that we accept to stop Algorithm 1 when the difference between scores is smaller than 0.1 (one tenth of the possible space) which is unreasonable in practice.

# E    Absolute Rank Error

In this section we present results based on the simulation settings in Section 5.1 with $n = 10,000$ and $\epsilon_f = 2$. Table 15 and Table 16 present the average absolute rank errors (over $H = 1,000$ runs) over different privacy budgets ($\epsilon_{CP}$) of the different private DP quantile approaches considered for Naive Bayes and Random Forest models respectively. In line with the results of Section 5.1, we observe that HistLap and P-COQS have the lowest errors over the different settings, with P-COQS showing generally better performance for smaller privacy budgets.

Table 15: Rank Errors for Naive Bayes (Private and Non-private)

| $\epsilon_{CP}$ | Method | Rank Error | | |
|---|---|---|---|---|
| | | ExponQ | LapHist | P-COQS |
| 0.1 | Non-private | 0.3372 (0.0154) | 0.0372 (0.0277) | 0.0316 (0.0251) |
| | Private | 0.1460 (0.0217) | 0.0231 (0.0189) | 0.0293 (0.0235) |
| 0.5 | Non-private | 0.1199 (0.0512) | 0.0084 (0.0066) | 0.0067 (0.0056) |
| | Private | 0.0813 (0.0222) | 0.0053 (0.0043) | 0.0067 (0.0055) |
| 1.0 | Non-private | 0.0574 (0.0230) | 0.0044 (0.0035) | 0.0040 (0.0031) |
| | Private | 0.0468 (0.0160) | 0.0030 (0.0023) | 0.0040 (0.0032) |
| 1.5 | Non-private | 0.0380 (0.0150) | 0.0032 (0.0025) | 0.0034 (0.0026) |
| | Private | 0.0326 (0.0117) | 0.0023 (0.0018) | 0.0033 (0.0025) |
| 3.0 | Non-private | 0.0194 (0.0078) | 0.0020 (0.0016) | 0.0028 (0.0019) |
| | Private | 0.0174 (0.0067) | 0.0017 (0.0013) | 0.0027 (0.0018) |
| 5.0 | Non-private | 0.0118 (0.0050) | 0.0016 (0.0013) | 0.0026 (0.0017) |
| | Private | 0.0109 (0.0045) | 0.0015 (0.0012) | 0.0026 (0.0016) |
| 10.0 | Non-private | 0.0061 (0.0030) | 0.0014 (0.0012) | 0.0026 (0.0016) |
| | Private | 0.0057 (0.0028) | 0.0015 (0.0012) | 0.0026 (0.0015) |

Table 16: Rank Errors for Random Forest (Private and Non-private)

| $\epsilon_{CP}$ | Method | Rank Error | | |
|---|---|---|---|---|
| | | ExponQ | LapHist | P-COQS |
| 0.1 | Non-private | 0.3615 (0.0139) | 0.0312 (0.0253) | 0.0237 (0.0211) |
| | Private | 0.3209 (0.0802) | 0.0182 (0.0133) | 0.0079 (0.0056) |
| 0.5 | Non-private | 0.1078 (0.0580) | 0.0074 (0.0054) | 0.0052 (0.0060) |
| | Private | 0.0871 (0.1125) | 0.0033 (0.0024) | 0.0037 (0.0021) |
| 1.0 | Non-private | 0.0467 (0.0204) | 0.0053 (0.0037) | 0.0028 (0.0045) |
| | Private | 0.0149 (0.0079) | 0.0023 (0.0017) | 0.0034 (0.0018) |
| 1.5 | Non-private | 0.0305 (0.0129) | 0.0048 (0.0032) | 0.0019 (0.0038) |
| | Private | 0.0099 (0.0053) | 0.0021 (0.0015) | 0.0032 (0.0016) |
| 3.0 | Non-private | 0.0160 (0.0073) | 0.0045 (0.0029) | 0.0009 (0.0028) |
| | Private | 0.0045 (0.0043) | 0.0020 (0.0013) | 0.0031 (0.0014) |
| 5.0 | Non-private | 0.0103 (0.0064) | 0.0044 (0.0028) | 0.0006 (0.0022) |
| | Private | 0.0019 (0.0032) | 0.0020 (0.0013) | 0.0031 (0.0013) |
| 10.0 | Non-private | 0.0044 (0.0052) | 0.0044 (0.0028) | 0.0003 (0.0017) |
| | Private | 0.0008 (0.0023) | 0.0019 (0.0013) | 0.0030 (0.0013) |

