# OpenReview forum: "Differentially Private Conformal Prediction via Quantile Binary Search"
_TMLR — Accepted by TMLR_

### Review · Reviewer_j1Zh · 2025-10-13

**Summary Of Contributions:**

This work proposes a general Differentially Private approach for Conformity Prediction via Quantile Search (P-COQS), which adapts an existing randomized binary search algorithm for computing DP quantiles.

Given a dataset, authors spilt it into a training dataset and a calibration dataset. Firstly, trained a model via training dataset, then compute an evaluation score for each data in the calibration dataset to evaluate the model (a lower score means this model more correctly predict this data). Finally, given a test data, it output a subset of the predictions, and this subset of candidates have lower evaluation scores.


1.	The “exchangeable” in this paper is confusing.
In the preliminary, authors say training data and calibrate data are exchangeable
However, in the proof of algorithm, authors treat calibrate data and training data are exchangeable. From the proof of Algorithm 1 and the second to the last sentence of Section 3.1, “Under the assumption of exchangeability of the data points, this set guarantees the coverage probability P(Ytest ∈ C(Xtest)) ≥ 1−α, where the probability is taken over the n + 1 data points (X1,Y1),...,(Xn,Yn),(Xtest,Ytest).”

2.	The research problem in this paper is less of meanings. Based on their analyses, there is no privacy issue of calibrate data here.
Authors consider calibrating dataset and the test data are exchangeable, actually, they treat them as IID, this is impossible. As the trainer could not get to know the test data, otherwise, he/she can just use this test data to train the model.

Even trying to fix and construct the application scenario, there are still some contradiction. From their algorithm, we can infer that, there should be 3 participators in the scenario. A has the training dataset and train the model, B has the test data, C (as a third party) evaluate the model (from A) to B. Note that, A and C could not be the same person as A could not know the test data. B and C could not be the same person as C should treat the test data privately. Then, there is also a contradiction. If C helps B to test the model, and B give C the data information, then there is no privacy issue.

3.	To output a subset of predictions, this in not useful. It is expected to output one exact prediction.

**Audience:**

No

**Audience Explanation:**

This paper (this version) is not suitable to TMLR.

**Claims And Evidence:**

No

**Claims Explanation:**

The research setting in preliminary is not consistent with the proof. For a test data should not be exchangeable with a calibrate data, so the proof theorem is not correct.

**Requested Changes:**

1.	Clarify the research question.
2.	The coverage guarantees are approximate, it do not guarantee exact (1-α)coverage; instead there is an error τ. Although the paper describes this as “controllable,” its quantitative analysis of that term under typical practical parameter ranges (e.g. n_cal,ρ,b−a) is weak, leaving readers unable to judge when the error may become unacceptable.
3.	The experiments of the manuscript were mainly compared with ExponQ (Angelopoulos et al., 2022), which is the appropriate preferred baseline, but other controls can also be supplemented to show the advantages and disadvantages more comprehensively.
4.	The article claims that P-COQS is significantly superior to ExponQ in computation (the average time difference is given in Table 4), but lacks an analysis of the algorithm complexity.
5.	The symbols in the text are not clearly indicated, for example, \delta.
6.	It is recommended to list both ρ (zCDP) and the converted (ε,δ) in the main text table, so that readers can visually assess the privacy intensity.
7.	The description at the beginning of the experiment section is too lengthy. It is suggested to polish the sentences to make them easier to read.
8.	In Theorem 1, Y_test should be Y

---

> ### Author Response · Authors · 2025-11-23
>
> We thank the reviewer for the detailed feedback and apologize if we did not clearly convey the context and the research question. Before addressing the main concerns, we note several improvements made beyond the reviewer comments:
> - we now use the notation $\Delta$ (instead of $\delta$) in P-COQS to avoid confusion with the ($\epsilon$, $\delta$) privacy parameter;
> - the main text now uses the Random Forest model (better illustrating the advantages of P-COQS, while Naive Bayes results remain similar);
> - we corrected an omission in the rank-error discussion (this does not change the qualitative conclusions);
> - we added a corollary showing how to adjust $\alpha$ to guarantee exact $(1-\alpha)$ coverage with probability $1-\beta$, mirroring the ExponQ guarantee with high probability.
>
> **1. Research question and exchangeability**
>
> Our work follows the standard split-conformal prediction framework, whose goal is uncertainty quantification via prediction sets with distribution-free coverage guarantees (and therefore does not aim for point predictions). The key assumption for this purpose is exchangeability of the full sequence $(X_i,Y_i)$, not that training data and test data are literally interchangeable in practice. More specifically, exchangeability is a probabilistic symmetry condition (e.g., i.i.d. data), not a data-handling requirement. Conformal prediction then proceeds as usual: train a model on the training split, compute non-conformity scores on the calibration split, and generate prediction sets for unseen test points.
>
> The motivation for privacy is that calibration scores or quantiles can leak information about individuals in the calibration split. While DP solutions exist for training the model, only ExponQ currently provides a DP mechanism for the quantile (calibration) step. Our goal is to study an alternative DP quantile mechanism which trades a small amount of theoretical tightness for empirical gains in coverage stability, efficiency, and computation.
>
> **2. Coverage guarantees**
>
> The reviewer is correct that P-COQS yields approximate (rather than exact) coverage. Our added corollary shows how adjusting the significance level can recover exact coverage with probability $1-\beta$. We are currently expanding the empirical study of rank errors across simulation regimes and will include these results in the revision. Despite approximate guarantees, our experiments show that the rank error remains small in practice, and empirical coverage remains close to target across multiple $\epsilon$, sample sizes, and choices of $\Delta$.
>
> **3. Additional baselines**
>
> We added a third DP quantile baseline: the histogram-Laplace / DP-CDF mechanism (denoted HistLap). While its conformal properties are not yet available (since it has been adapted for this work), it is a natural alternative. Across experiments, P-COQS matches or outperforms HistLap, while requiring fewer tuning constraints: P-COQS only requires a small $\Delta$, whereas both ExponQ and HistLap require selecting a histogram/bin width, which is known to be delicate and privacy-budget dependent.
>
> **4. Algorithmic complexity**
>
> We added a proposition giving the computational complexity of P-COQS (linear in the number of binary-search steps and logarithmic in the score discretization resolution). Comparable analyses are not available for ExponQ or HistLap due to their histogram-optimization procedures, but empirically their runtimes are similar except for the additional bin-search overhead.
>
> **5. Notation**
>
> All symbols (including the revised $\Delta$) have been clarified and checked for consistency to the best of the authors' understanding.
>
> **6. Privacy reporting**
>
> We now list both $\rho$ (zCDP) and the corresponding ($\epsilon$, $\delta$) parameters where relevant in the experiments, as suggested by the reviewer.
>
> **7. Experimental section**
>
> We streamlined the introductory text of the experiments as much as possible without losing essential information. We however welcome more precise suggestions from the reviewer on this point and would be happy to address them accordingly.
>
> **8. Notation for test labels**
>
> We retain $Y_{test}$ for clarity since replacing it with Y could create ambiguity about whether the statement refers to calibration or training labels as well.
>
> **Overall:**
> We hope that the changes in the current version of the revised manuscript (to which some results are still to be added) have contributed to (partially) addressing some of the reviewer's concerns and can support a productive discussion towards strengthening its message and contributions.

---

### Review · Reviewer_7vqy · 2025-10-16

**Summary Of Contributions:**

The paper introduces P-COQS, a differentially private conformal prediction method that releases a DP quantile of calibration nonconformity scores via a fixed-depth binary search over a *known score range* [a,b] (e.g., the observed min/max of calibration scores or user-specified bounds) with resolution delta, using a noisy range count (NoisyRC) at each step; noise is calibrated so the search satisfies $\rho$‑zCDP. The theory provides a high-probability rank‑error bound for the privatized quantile and, under exchangeability, an approximate coverage band for the resulting conformal sets around the nominal $(1 − \alpha)$. Experiments on simulations and three vision benchmarks (CIFAR‑10, ImageNet, CoronaHack), with both non‑DP and DP base models, report near‑nominal coverage, smaller prediction sets than ExponQ, and orders‑of‑magnitude faster runtime.

**Additional Comments:**

The review has been modified after the rebuttal to reflect that the under-substantiated claims have now been addressed.

**Audience:**

Yes

**Audience Explanation:**

I believe TMLR's audience would be interested in the paper as it provides

- **Privacy-aware uncertainty quantification:** A simple, composable mechanism that privatizes the conformal quantile with clear zCDP accounting.

- **Applied ML in sensitive domains:** The runtime advantage and smaller sets at comparable coverage are attractive for clinical and large-scale vision deployments.

- **Theory–practice bridge:** An explicit rank-error to coverage band mapping that can be budgeted alongside DP-SGD training privacy.

**Claims And Evidence:**

Yes

**Claims Explanation:**

Overall, the paper’s claims are substantiated for the most part, with several focused gaps that could be addressed.

**(C1) Claim (Sec. 4):** *The paper proposes a private quantile mechanism for continuous scores via binary search (P-COQS).*
**Evidence:** Algorithm 1 adapts a private quantile search to real-valued scores by searching over an interval ([a,b]) at resolution ($\delta$) for a fixed number of steps ($N=\lceil\log_2((b-a)/\delta)\rceil$). Each step calls a noisy range count to estimate (#$\{s\le \text{mid}\}$) with Gaussian noise; the final midpoint is the DP quantile ($q_{\mathrm{DP}}$). Algorithm 2 forms the conformal set using ($q_{\mathrm{DP}}$).

**(C2) Theoretical claim (Sec. 3.2–4):** *The DP quantile search satisfies rho‑zCDP and induces a high‑probability rank‑error tau, which together yield an approximate coverage band for the resulting DP conformal sets.*
**Evidence:** Gaussian noise per range‑count query is scaled to the fixed iteration budget $N$, so each call uses $\rho/N$ and the search composes additively to $\rho$‑zCDP. With probability at least $1−\beta$, the privatized quantile’s rank deviates from the non‑private rank by at most $\tau$ (explicit tail bound provided). Under exchangeability, this rank‑error window implies the coverage band, and the text discusses deterministic tie handling.

**(C3) Claim (Sec. 5):** *Across benchmarks, P-COQS attains near-nominal coverage, is more stable across privacy budgets and dataset regimes, and is substantially faster than ExponQ.*
**Evidence:** Section 5 (simulations and CIFAR-10, ImageNet, CoronaHack) reports near-target coverage with smaller, more stable prediction sets than ExponQ, and orders-of-magnitude runtime speedups (e.g., $\approx$0.007 s vs $\approx$0.75 s in the NB setting), supporting the central empirical claim.

## Claims that are undersubstantiated

**(U1) Exact claim:** *P-COQS achieves favorable efficiency at comparable coverage across settings.*
**Provided:** Head-to-head comparisons with ExponQ on simulations and three benchmarks show smaller set sizes for P-COQS at near-nominal coverage; CIFAR-10 also includes a non-private CP baseline (“Standard”).
**Undersubstantiated & why it's required:** Comparisons are limited to ExponQ as the DP-quantile baseline. Alternative DP quantile estimators (e.g., smooth-sensitivity quantiles; histogram + Laplace / DP-CDF; direct DP-quantile mechanisms) are not included. Adding one such baseline would demonstrate that the efficiency advantage is not specific to ExponQ’s inflation strategy, improving external validity of the claim. _(Representative references provided in the References section.)_

**(U2) Exact claim:** *The continuous-score adaptation (grid ($[a,b],\delta$)) is robust in practice.*
**Provided:** Theory specifies iteration budget ($N=\lceil\log_2((b-a)/\delta)\rceil$) and a rank-error bound that scales with ($u=(b-a)/\delta$).
**Undersubstantiated & why it's required:** No ablation shows sensitivity of coverage/set size to bounds ([a,b]) or resolution ($\delta$). Since the error term depends on ($u$), empirical plots vs ($[a,b],\delta$) (and observed ($\tau$) percentiles) are needed to establish practical robustness and to guide practitioners in choosing these knobs.

**(U3) Exact claim:** *The approximate coverage band is adequate in practice.*
**Provided:** A two-sided coverage band ($1-\alpha\pm O((\tau+1)/(n_{\mathrm{cal}}+1))$) with discussion of ties and approximation sources.
**Undersubstantiated & why it's required:** The magnitude of ($\tau$) under realistic settings is not reported (e.g., empirical distribution of rank error across runs). Reporting ($\tau$) percentiles alongside coverage would quantify the tightness of the band and clarify when small calibration sizes materially affect coverage.

---
# References
[1] Nissim, K., Raskhodnikova, S., & Smith, A. (2007). Smooth sensitivity and sampling in private data analysis. STOC ’07.
[2] Bun, M., Nissim, K., Stemmer, U., & Steinke, T. (2019). Private algorithms for smooth sensitivity and mean estimation. NeurIPS.
[3] Wasserman, L., & Zhou, S. (2010). A statistical framework for differential privacy. JASA, 105(489), 375–389. (DP histograms/DP-CDF; quantiles via CDF.)
[4] Li, C., Hay, M., Miklau, G., & Wang, Y. (2015). Differentially private histogram publication for dynamic datasets. VLDB J.
[5] Kaplan, H., Mansour, Y., & Stemmer, U. (2022). Differentially Private Approximate Quantiles. ICML 2022.

**Requested Changes:**

The following changes will help substantiate all the claims made and make the paper read better.

- **[Definite] Adding one DP-quantile baseline beyond ExponQ.** Including a smooth-sensitivity or histogram + Laplace / DP-CDF quantile to show that P-COQS’s efficiency gains are not baseline-specific. _(See References in Section above.)_

- **[Definite] Ablating ($[a,b]$), ($\delta$), and ($\beta$).** Reporting coverage and mean set size vs these knobs and include empirical percentiles of rank error ($\tau$) to operationalize the theory and guide practice.

- **[Minor if not modify the claim] Slice-wise / shift robustness.** Adding a small Mondrian (per-class) analysis or a covariate-shift stress test to assess coverage beyond the marginal guarantee.

- **[Optional] Reporting CP-step privacy also in (($\varepsilon,\delta$)).** Providing the standard zCDP to DP conversion so the CP privacy budget can be compared with DP-SGD training budgets.

- **[Optional] Clarifying implementation details.** Stating the tie-handling rule used in experiments (deterministic “($\le$)” vs randomized), and make Algorithm-1 midpoint/return lines explicit for readability.

- **[Optional] CIFAR-10 reminder.** In Sec. 5.2.1, including a one-line reminder that “Standard” refers to non-private split CP on the same splits, for quick orientation.

---

> ### Author Response · Authors · 2025-11-23
>
> We sincerely thank the reviewer for the exceptionally thorough and constructive review. We greatly appreciate the detailed suggestions, which helped us strengthen both the theoretical and empirical components of the paper. Before addressing the specific points, we note several improvements to the revised manuscript:
> - We replaced $\delta$ with $\Delta$ in P-COQS to avoid conflict with the ($\epsilon$, $\delta$) privacy parameter.
> - Random Forest is now the main example, since it highlights the differences between DP quantile mechanisms more clearly (Naive Bayes results remain similar).
> - We corrected an omission in the rank-error expression. This does not alter the qualitative guarantees but clarifies the dependence on the final search interval.
> - We added a corollary following Theorem 1 that enables exact $(1-\alpha)$ coverage with probability $1-\beta$ by adjusting the significance level.
>
> Below we address the reviewer’s unsubstantiated-claim points.
>
> **U1 — Additional DP-quantile baselines**
>
> We agree that including another DP quantile mechanism helps separate the advantages of P-COQS from ExponQ’s inflation strategy. Following the reviewer’s suggestion, we implemented the histogram + Laplace / DP-CDF mechanism (denoted HistLap) and added it to all simulation studies.
>
> Across the various regimes:
> - P-COQS outperforms both ExponQ and HistLap for the Random Forest model.
> - For Naive Bayes, P-COQS typically matches HistLap and outperforms ExponQ, with P-COQS regaining the advantage for small ε or small n.
>
> Importantly, HistLap requires choosing a bin width, which significantly affects its utility. P-COQS only requires choosing $\Delta$, and after correcting the rank-error expression, the analysis shows that making $\Delta$ small is generally recommended: the privacy noise grows only logarithmically in $(b-a)/\Delta$, whereas histogram-based errors depend more sensitively on binning. We added a remark in the experiments section summarizing these practical implications.
>
> **U2 — Ablation over $u = (b-a)/\Delta$ through $\Delta$**
>
> We added an ablation study examining the effect of $\Delta$ in the range $10^{-20}$ to $10^{-1}$ under fixed bounds [0,1]. Indeed, given that the bounds on the data/scores are something that are generally defined *a-priori* (before seeing the data) or are natural bounds (such as the case for classification) we focused on the choice of $\Delta$ (the main user-defined parameter) since this can determine the size of $u$, having arbitrarily fixed $(b-a)$ (see added Remark 1 on bounds in the revised manuscript). The results (Appendix D) show that P-COQS is stable across essentially all practical values. Only $\Delta = 10^{-1}$ produces noticeable degradation (over-coverage and larger sets), which is expected given that this choice coarsens the search substantially. Following our theoretical correction, the guidance is now explicit: choose \Delta as small as computationally convenient.
>
> **U3 — Practical adequacy of the approximate coverage band**
>
> We are planning to add empirical rank-error summaries across simulation settings (which will be discussed where relevant). Given the current results for coverage (and other metrics), we expect these to show that $\tau$ remains reasonably small even under strict privacy and small calibration sets (which indeed matches the empirical coverage stability observed in Section 5).
>
> **Other suggested changes**
> - Slice-wise / shift robustness: we clarified that our claims pertain to marginal coverage under exchangeability. Following the reviewer’s suggestion, we added a remark after Theorem 1 explicitly stating that extensions to group-conditional or covariate-shift coverage are outside the scope of the current work.
> - Reporting CP-step privacy in ($\epsilon$, $\delta$): we now include the zCDP–DP conversion in the experiments for easier comparison with DP-SGD training budgets.
> - Implementation details: ties did not occur in our experiments (continuous scores), but we clarified this. We are happy to update Algorithm 1 further once the reviewer specifies what additional midpoint/return details would be preferable.
> - CIFAR-10 reminder: we added the requested one-sentence clarification that “Standard’’ refers to non-private split conformal prediction using the same splits.
>
> We thank the reviewer again for the highly constructive feedback, which helped us significantly strengthen the manuscript.

---

> ### Comment · Reviewer_7vqy · 2025-11-24
>
> Thank you for those clarifications and changes. I am happy with the experiments performed to substantiate U1 and U2.
>
> I also understand that experiments to substantiate U3 are still running due to time constraints. Please do include this in the camera-ready paper, if accepted.
>
> If it is not possible to substantiate this claim by then, please rephrase the wording to reflect only what is substantiated.
>
> That being said, since my other concerns have been answered, I will modify my review to reflect that the claims made in the paper are now substantiated.

---

> > ### Author Response · Authors · 2025-12-01
> >
> > We greatly thank the reviewer for their response and can confirm that we have now added results on the rank error which are in line with those of the current results on coverage and other metrics. We thank them again for their very constructive feedback and for allowing us to improve our work.

---

### Review · Reviewer_QF3c · 2025-11-17

**Summary Of Contributions:**

- The authors initially describe the challenges of deploying conformal prediction (CP) as a truly private solution for generating distribution-free samples, noting the possibility of a privacy-leaking calibration step.
- Note that CP is often used in high-sensitivity fields where, along with privacy guarantees where high levels of actual coverage (not via just simply data distribution) are necessary.
- The authors mention that the calibration step in CP can leak information since the quantile of calibration non-conformity scores depends on the underlying individual's data.
- The paper notes that existing exponential methods are not sufficient due to the high cost of binning, tuning, and possibly inflated quantiles, leading to larger-than-required prediction sets.
- The authors then propose P-COQS, which enables a differentially private binary search for identifying the quantile on individual conformal scores with better accuracy than existing methods.
- Errors due to DP are quantified both theoretically and via experiments.

**Audience:**

Yes

**Audience Explanation:**

The work focuses on differentially private metric generation to generate distribution-free prediction samples. Statistical findings can be useful in systems with limited data with high levels of privatization.

**Claims And Evidence:**

Yes

**Claims Explanation:**

Yes, the core claims of the paper can be categorized as per the following,

First, the theoretical bounds, as recognized by the privacy bound for Algorithm 1 (this is by definition), rank error probabilistic bound from Proposition 1 (ensuring probabilistic accuracy), and the subsequent coverage guarantee in Theorem 1, are responsible for ensuring privacy-accuracy guarantees.

Second, the empirical results show a better coverage than existing SOTA methods for the low-$\epsilon$ regime.

However, the following are some limitations that need to be addressed in this work,
1. The authors describe the lower and upper bounds of the scores $(a, b)$. How are these readily available? And are they fully disclosed or only disclosed under DP? Otherwise, the value of $q$ might leak information.
2. Is it possible to compare the method to other possible alternatives? Like DP histogram releases, the release of quantile under already privatized data / privatized model, other uncertainty techniques that might work with DP?
3. Possibility to explore membership inference attack-based verification for quantile releases that can strengthen empirical results?
4. How does the method work for continuous targets, such as in the case of regression?

**Requested Changes:**

Overall, the paper is well-written and mostly justifies its claims. However, I have identified 4 important limitations of the work above that can be addressed for significant improvements.

---

> ### Author Response · Authors · 2025-11-23
>
> We thank the reviewer for the positive and encouraging assessment of our work, as well as for highlighting several limitations that helped us improve the manuscript. Before addressing the specific points, we note a few general changes implemented in response to all reviewers:
> - $\delta$ in the algorithm has been changed to $\Delta$ to avoid conflict with the ($\epsilon$,$\delta$) definition of DP.
> - Random Forest is now the main running example, as it highlights the differences among DP quantile mechanisms more clearly (Naive Bayes results remain consistent).
> - We corrected an omission in the rank-error expression: the final error also depends on the number of calibration scores within the terminal search interval $\Delta$. This does not affect the overall theoretical guarantees but clarifies the dependence on user-chosen resolution.
> - Following Theorem 1, we added a corollary allowing the user to adjust the nominal $\alpha$ to guarantee exact $(1-\alpha)$ coverage with probability $1-\beta$.
>
> Below we address the reviewer’s four main points.
>
> **1. On the choice and disclosure of bounds [a,b]**
>
> As in many DP applications, defining bounds requires either (a) knowledge external to the data, or (b) spending privacy budget to estimate them privately. If bounds were derived non-privately from the sample, releasing a DP quantile could indeed leak information.
>
> For this reason, bounds must be:
> - known a priori (e.g., classification non-conformity scores lie in [0,1]),
> - or natural from preprocessing (e.g., standardized residuals),
> - or obtained via a DP mechanism if they depend on the data.
>
> We added a remark (Remark 1) clarifying these options. In our experiments, all scores naturally lie in [0,1].
>
> **2. Comparison with other DP quantile mechanisms**
>
> We agree that comparing only to ExponQ is limiting. Following the reviewer’s suggestion (also raised by the other reviewers), we added a DP baseline based on histogram + Laplace / DP-CDF estimation (denoted HistLap). HistLap is now included throughout the simulation studies.
>
> Empirically:
> - P-COQS generally outperforms both ExponQ and HistLap for Random Forest.
> - For Naive Bayes, P-COQS is comparable to HistLap but consistently better than ExponQ; under small privacy budgets or small samples, P-COQS regains a clear advantage.
>
> Theoretically, the corrected rank-error expression highlights a key difference: P-COQS benefits from choosing $\Delta$ extremely small (privacy cost grows only logarithmically), whereas histogram methods must choose an optimal bin width, which directly affects accuracy and variance. We summarised these points in an added remark (Remark 8).
>
> 3. Membership-inference attacks (MIAs)
>
> We appreciate this suggestion. In general, MIAs test empirical privacy leakage for non-DP mechanisms. Since P-COQS releases a quantile through a provably $\rho$-zCDP mechanism, any adversary’s advantage in a membership inference attack is already upper-bounded by the DP guarantee: a MIA cannot violate or exceed the DP bound.
>
> Nevertheless, MIAs can still be used as a diagnostic to confirm that empirical leakage is consistent with DP theory. We added a remark (Remark 5) explaining this perspective and identifying MIAs as a promising complementary evaluation direction for practitioners, while clarifying why they do not provide additional formal guarantees beyond DP itself.
>
> **4. Extension to continuous targets (regression)**
>
> The method applies directly to regression settings. The only change is the choice of the score bounds [a,b], since the binary search is performed over the range of the non-conformity scores. Our current experiments already involve continuous non-conformity scores (derived from predicted probabilities in [0,1]), and PrivQuant was adapted precisely to handle continuous values rather than discrete ranks.
>
> We focused on classification tasks to align with the experimental setting of ExponQ, but the algorithm requires no structural change for continuous targets. We added clarifying text in the revised manuscript.
>
> We again thank the reviewer for the thoughtful comments, which substantially improved the clarity and completeness of the paper.

---

### Decision · Action_Editor_pV5j · 2026-01-13

**Recommendation:** Accept as is

**Audience:**

Yes

**Audience Explanation:**

I think this is very well fitting for TMLR: private confidence intervals for DP trained models via conformal prediction (meaning private also for the calibration data).

The paper meets the expectations for publication in TMLR. Two out of three reviewers support acceptance and their comments have been addressed during the review process. Also, I think the comments of the third reviewr (j1Zh) have been sufficiently clarified by the authors.

**Claims And Evidence:**

Yes

**Claims Explanation:**

After reviewing the discussion and responses, my understanding is that all claims are correct. The paper also demonstrates novelty although applying an existing DP quantile search method. The authors have adequately addresses most reviewer concerns. To the best of my understanding, Reviewer j1Zh's critique appears to stem from some misunderstandings of the conformal prediction techniques and its assumptions, which the authors have clarified.